

# Geochronological reconstruction of the glacial evolution in the Ésera valley (Central Pyrenees) during the last deglaciation

Ixeia Vidaller[1], Toshiyuki Fujioka[2], Juan Ignacio López-Moreno[1], Ana Moreno[1], ASTER Team[3,*]

[1] Instituto Pirenaico de Ecología, Consejo Superior de Investigaciones Científicas (IPE-CSIC), Zaragoza, Spain

[2] Centro Nacional de Investigación sobre la Evolución Humana (CENIEH), Burgos, Spain

[3] CNRS, Aix Marseille Univ, IRD, INRAE, CEREGE, Aix-en-Provence, France

[*] Aster Team: A full list of authors appears at the end of the paper.

*Correspondence to*: Ixeia Vidaller (ixeia@ipe.csic.es)

**Abstract.** The last deglaciation period in the Pyrenees was distinguished by intricate glacier dynamics, encompassing a
multitude of advances and rapid glacier retreats that did not always align with the fluctuations observed in other European glaciers. The Ésera valley, located in the Central Pyrenees (northern Spain), provides a distinctive opportunity to reconstruct past climate in high-mountain regions during the last deglaciation period. Previous studies of glacial evolution in this area have employed a variety of methods, including the analysis of glacial lake sediments and detailed geomorphological studies of glacial landforms. This paper presents measurements of cosmogenic $^{10}$Be exposure ages from glacial deposits and a polished
bedrock surface in the Ésera valley, together with calculations of the equilibrium line altitude (ELA), with the objective of reconstructing the evolution of the Ésera glacier and the associated environmental implications during the last deglaciation. Following the Pyrenean Last Glacial Maximum, at approximately 75 ka in the Ésera valley, the Ésera glacier commenced a period of retreat during the Marine Isotopic Stage (MIS) 3, reaching a point of stabilisation at approximately 47 ka at the location of the Pllan d'Están proglacial lake. Subsequently, a new glacial advance resulted forming the Llanos del Hospital
moraine (~16 ka), a glacial deposit located a lower altitude in the valley than Pllan d'Están lake. During that time interval, we suggest that sediment deposition at Pllan d'Están took place in a subglacial environment. Following the conclusion of the Oldest Dryas period (~16 ka) and continuing into the Early Holocene, the Ésera glacier underwent a rapid retreat. The Little Ice Age (LIA) represented the last cold period documented in the Ésera valley, after which the glacier has exhibited a persistent retreatment. The ELA analyses indicate that the temperature in the Ésera valley increased by 3.6 ± 0.45 ℃ over the past 16 ka,
which resulted in the retreat of the glacier front from 1750 metres above sea level (m a.s.l.) to 3000 m a.s.l.

**Keywords:** cosmogenic exposure dating, Pyrenees, last deglaciation, glaciers, glacial paleogeography, paleoELAs.

## 1 Introduction

Glaciers are highly effective indicators of climate variability due to their sensitivity to climate change, particularly in relation to temperature and precipitation. Consequently, their thickness and surface variations are regarded as one of the most
informative proxies for global warming in mountainous regions (e.g. Beniston et al., 2018; Braithwaite and Hughes, 2020). A





number of approaches have been developed for the reconstruction of past glacier evolution, which are typically based on the analysis of glacial geomorphologies, including till or moraine deposits and polished bedrock. One approach, arguably among the most widely used methods over the past couple of decades, is to date glacial landforms with cosmogenic isotopes. The application of cosmogenic exposure dating has become a fundamental technique for the reconstruction of glacial history,

including the advance and retreat of glaciers (Ivy-Ochs and Briner, 2014; Balco, 2020; see references therein). Despite the discontinuous nature of glacial deposits, the dating of erratic boulders, moraines, or polished bedrock via exposure dating is a commonly used method for studying the dynamics of glaciers and the climate changes that occurred during the Last Glacial Cycle (LGC), which spanned from approximately 120 to 11.7 ka. When possible, that discontinuous information is complemented by the study of lacustrine sediments associated to the glacier dynamics (e.g. Moreno et al., 2010).

The Equilibrium Line Altitude (ELA) is also a highly effective method for inferring past climate variations, identifying the altitudinal boundary between the accumulation zone (where processes favouring snow deposition are dominant) and the ablation zone (where the glacier loses ice mass). The ELA is highly sensitive to climatic changes that modify the extension of the accumulation and ablation zones. This allows the calculation of past environmental variability through the variations in reconstructed paleoELAs (Sissons and Sutherland, 1976; Sutherland, 1984; Dahl and Nesje, 1992). In order to determine the

paleoELA, it is necessary to have accurate knowledge of the glacier extension at a given time period. This is usually achieved through the dating of glacial landforms, such as till and polished bedrocks. Despite the challenges associated with interpreting ELAs in the context of climate change, particularly the identification and dating of geomorphological features that are not always well preserved, this approach remains a valuable method for understanding past thermal changes in mountainous environments (Pellitero et al., 2019).

The Pyrenees currently host the southernmost glaciers in Europe (Grunewald and Scheithauer, 2010), which are classified as very small due to their limited extension (<0.5 km²; Huss and Fischer, 2016). However, during the Pyrenean Last Glacial Maximum (PLGM) period, which occurred at approximately 60-70 ka corresponding to Marine Isotopic Stage (MIS) 4, as documented by Lewis et al. (2009), glaciers covered a significant portion of the region above 800 m a.s.l. Consequently, they constituted a prominent element of landscape modeling in the Mediterranean area, as evidenced by the findings of Oliva et al.

(2019). Subsequent to the PLGM and throughout the LGC, a number of glacial phases have occurred. Some are better preserved as a later stage of the PLGM during MIS 4, for example in the Ara valley (Bartolomé et al., 2021) or in the Gállego valley (Lewis et al., 2009). Other phases are less well preserved and are the subject of some controversy or inconsistency with other European regions. For example, the Last Glacial Maximum (LGM) was defined as an interval centred on 21 ka (Mix et al., 2001), characterised by a global increase in glacier extent (Hughes et al., 2013). However, in the Pyrenees and other Iberian

mountains (Palacios et al., 2011, 2012, 2016), this period was also characterised by low temperatures but high aridity, which seemingly prevented a significant growth of glaciers, as occurred in many Mediterranean mountains (Batbaatar et al., 2018; Allard et al., 2021). Since just few moraines were dated from this period in certain Eastern Pyrenean valleys (Noguera-Ribagorzana valley; Delmas, 2015; Pallàs et al., 2006), numerous previous studies have proposed an earlier deglaciation in the





Pyrenees that would have commenced shortly after the PLGM (Lewis et al., 2009; García-Ruiz et al., 2012; García-Ruiz et al., 2013; Oliva et al., 2019).

The Ésera valley is currently the most glaciated valley in the Pyrenees (Rico et al., 2017; Vidaller et al., 2021) due to its geographic location, which may result in a complex influence from both the Mediterranean and Atlantic climates. In the context of current climate change, measurements of the extent and thickness of the modern glaciers (namely, Aneto, Maladeta and Tempestades glaciers) were carried out to investigate the recent evolution of the glaciers in the valley (Mora et al., 2006; Pastor Argüello, 2013; Jiménez-Vaquero, 2016; Rico et al., 2017; Campos et al., 2021; Vidaller et al., 2021, 2023). In contrast, there is a paucity of studies that have focused on the past extent of the glaciers. Indeed, there are only two examples that have dated moraines and polished bedrock from the Maladeta massif via cosmogenic exposure dating (Crest et al., 2017; Reixach, 2022). The use of different geomorphological maps (1:50,000) of the Maladeta massif may prove beneficial in determining the evolution of the glaciers in the Ésera valley (Martínez de Pisón, 1989, 1990; Bordonau, 1992, 1993; García-Ruiz et al., 1992; Chueca-Cía and Julián-Andrés, 2008). In a recent study, Vidaller et al. (2024a) produced a detailed geomorphological map of the Maladeta massif, with the objective of facilitating the identification of glacier deposits and landforms at a higher resolution (1:15,000). The sediments from the two paleolakes in the valley have also been the subject of study. The glaciolacustrine rhythmites of Barrancs lake (2360 m a.s.l.) have been observed to cover the last 3 ka BP (Copons and Bordonau, 1997; Copons et al., 1997). On the other hand, the sedimentary sequence of the Pllan d'Están paleolake (1840 m a.s.l.) evidenced main climatic variations that have occurred over the past 47 ka (Vidaller et al., 2024b).

The region offers a distinctive opportunity for glacial and paleoclimate research, as it allows for the investigation of the last deglaciation through a range of methods in a single area. These include the use of cosmogenic dating of moraines, proglacial lake sediments (Vidaller et al., 2024b) and modern glacier extent (Vidaller et al., 2021, 2023). In this paper, we present the measurements of cosmogenic $^{10}$Be from granitic boulders from moraines and polished bedrock from the Ésera valley, together with ELA calculations, with the aim of reconstructing the chronological sequence of the deglaciation of the Ésera glacier. Subsequently, we examine the paleoenvironmental conditions and history of glacial advance and retreat in the study area, integrating our cosmogenic exposure ages with the preceding sedimentary study of the Pllan d'Están paleolake. Additionally, paleotemperature estimation from different deglaciation stages in the Ésera valley will facilitate a more quantitative determination of the paleoenvironmental condition.

## 2 Study area

The Ésera valley is situated in the Central Pyrenees in northern Spain (Fig. 1a). At the headwaters, the southernmost slope of the valley corresponds to the northern slope of the Maladeta massif, which is home to the largest concentration of the last remaining glaciers in the Pyrenees. The massif comprises over 40 peaks exceeding 3000 m a.s.l., including some of the highest in the Pyrenees, such as the Aneto peak, which reaches 3404 m a.s.l. A distinctive feature of this valley is its circuitous route



around the Maladeta massif, which results in a notable shift in orientation from north-northwest to south-southeast to north-northwest in a span of less than 10 km.

In geological terms, the study area is located at the northern boundary of the tardihercinic granitic batholith of the Maladeta massif, which is one of the batholiths that constitute the Axial Zone of the Pyrenees, and it is situated above the Paleozoic sedimentary and metamorphic units. The lower areas of the valley are characterised by the presence of Devonian limestones,

which are intercalated with Carboniferous shales and quartzites. The northern slope of the valley has been subject to influence from both the Hercynian and Alpine orogenies, as well as the outcropping of Ordovician shales, quartzites and sandstones. The northern slope does not attain the same elevations as the southern slope. The upper part of the Maladeta massif is composed of granite and granodiorite of the batholith (Ríos-Aragüés et al., 2002; García-Sansegundo et al., 2013) (Fig. 1b).

Attending to the geomorphological characteristics, the Maladeta and Ésera landscape has been mostly modelled by the action

of glaciers along the Quaternary (Martínez de Pisón, 1989, 1990; García-Ruiz et al., 1992; Copons and Bordonau, 1997). The glacial and periglacial modelling is dominating in the massif and especially visible in granites, while limestones often show karstic landforms (Vidaller et al., 2024a). The variation in lithologies from granite at the top and limestone at the bottom of the valley allows us to identify the source area of the glacial deposits (Fig 1c), which in turn enables the reconstruction of paleoglacial activities in the region. The prevalence of limestone in extensive areas has resulted in the formation of diverse

karst features, including sinkholes, poljes and ponors, which have a significant impact on the hydrological system. Numerous depressions within the valley have been filled with glacial or lacustrine sediments (and may also have undergone erosion due to glacial advances), which, in conjunction with glacially-transported deposits (e.g. till, moraines), serve as a valuable archive of past climate records.





**Figure 1: Location of the Ésera valley and the Maladeta massif. (a) Location of the study area in the European context. Pink pin indicates the location of the Maladeta massif in the Pyrenees. Hillshade obtained from the European Environment Agency. (b) Major geological characteristics of the Maladeta massif and the Ésera valley (modified from Vidaller et al., 2024a). Names of the main locations considered in this work are indicated. Viewpoint of Figure 1c is also marked. Hillshade obtained from Spanish National Geographic Institute. (c) Overview of the headwaters of Ésera valley and northern slope of the Maladeta massif with the key locations mentioned in this paper. The picture is taken by looking to the south (note the position of the North as indicated by the arrow).**

## 3 Methods

### 3.1 Terrestrial cosmogenic nuclide exposure dating

To reconstruct the evolution of the Ésera glacier, a total of 14 samples were collected for cosmogenic exposure dating in order to determine the timing and position of glaciers during the last glacial cycle in the Ésera valley. The samples included 13 granitic erratic or moraine boulders and one from a polished quartzite bedrock (Fig. 2, Table 1; see *Supplementary Table S1*



for field data). In order to select the most appropriate samples, the detailed geomorphological map of the Maladeta massif (Vidaller et al., 2024a) was consulted. Granite boulders large enough to avoid post-depositional movements due to glacier melting were considered. In addition, special attention was paid to ensure that the boulders had a flat surface to ensure the highest solar incidence. The samples were collected from a range of elevations, with ten originating from the middle and lower

reaches of the Ésera valley at elevations between 1744 and 1995 m a.s.l., and the remaining four from the upper part of the valley in the Maladeta massif at elevations between 2039 and 2634 m a.s.l. (Fig. 2). Two samples (PDE-1 and -2) were collected in the vicinity of Pllan d'Están paleolake and are directly comparable to the chronologies reconstructed from the lacustrine sequence in our previous study (Vidaller et al., 2024b).

The samples were processed at the Cosmogenic Nuclide Dating Laboratory in CENIEH, Burgos, Spain, and nuclide

measurements were carried out at ASTER, CEREGE, Aix-en-Provence, France (see *Supplementary Table S2* for details of sample preparation and AMS data). The sample preparation protocol was modified from previously established procedures (Kohl and Nishiizumi, 1992; Fujioka et al., 2022). In summary, the rock samples were crushed and sieved into fractions of grain size between 0.25 and 0.50 mm, followed by density separations at 2.62 and 2.68 g cm$^{-3}$, leaching by aqua regia and finally etching by diluted HF to obtain pure quartz. A quantity of between 12 and 30 g of the pure quartz was spiked with

approximately 0.25 to 0.27 mg of $^9$Be and dissolved in concentrated HF. Subsequent to dissolution of the quartz, small aliquots were obtained for analysis of aluminium by ICP-OES at CENIEH. Beryllium (and aluminium for a bedrock sample, PDE-3) was isolated from unwanted cations via sequential anion and cation exchange chromatography. Subsequently, hydroxides were precipitated with Suprapur NH$_3$ (aq.). The Be- (and Al-) hydroxides were then transported to ASTER, where they underwent further processing to become BeO (and Al$_2$O$_3$). Finally, they were compressed into Cu cathodes with Nb powder (and Ag

powder for Al$_2$O$_3$) for AMS measurements.

AMS $^{10}$Be/$^9$Be (and $^{26}$Al/$^{27}$Al) measurements were conducted using the ASTER 5MV accelerator at the Centre de Recherche en Écologie et Gestion de l'Environnement (CEREGE) (Arnold et al., 2013). The $^{10}$Be/$^9$Be measurements were normalised against the STD-11 standard, with a nominal ratio of 1.191x10$^{-11}$, while the $^{26}$Al/$^{27}$Al measurements were normalised against SM-Al-1. One standard was used, with a nominal ratio of 7.401x10$^{-12}$ (Merchel and Bremser, 2004; Arnold et al., 2010;

Braucher et al., 2015). The measurements of chemistry procedural blanks indicate a ratio of 2.6-4.0x10$^{-15}$ for $^{10}$Be/$^9$Be and 1.6-5.5x10$^{-15}$ for $^{26}$Al/$^{27}$Al. These values are typically ~2% and ~1% of the measured ratios for the samples, respectively. The $^{10}$Be (and $^{26}$Al) concentrations were calculated from the measured ratios after standard normalisation and blank correction. The final errors were calculated from AMS uncertainties, including counting statistics, standard reproducibility, the error on the standard nominal ratio, and blank correction. Additionally, a 1% error in the $^9$Be carrier concentration was considered, or in the case of

$^{26}$Al, a 3% error in the intrinsic $^{27}$Al measurement by ICP-OES. These errors were calculated in quadrature. The surface exposure ages were calculated using version 3 of the CRONUS-Earth online calculator (http://hess.ess.washington.edu/, accessed February 2024; Balco et al., 2008), employing the LSDn scaling scheme (Lifton et al., 2014) (see *Supplementary Table S3*). The snow shielding factor was calculated in accordance with the methodology outlined by Gosse and Phillips (2001), with an assumed average snow density of 0.4 g cm$^{-3}$ and the utilisation of monthly snow thickness data from the Ésera



valley region between 2007 and 2023 (AEMET database). The calculated age estimates are considered to be the most accurate when the erosion rate of 3 mm ka$^{-1}$ and the snow shielding correction are applied. It should be noted that PDE-1 and PDE-2 are situated in the lowest part of the valley (Fig. 2), where snow accumulation is likely to have been greater than the average. Therefore, for these two samples, the calculated age estimates are the most precise when the snow shielding factors are calculated without considering the boulder heights.

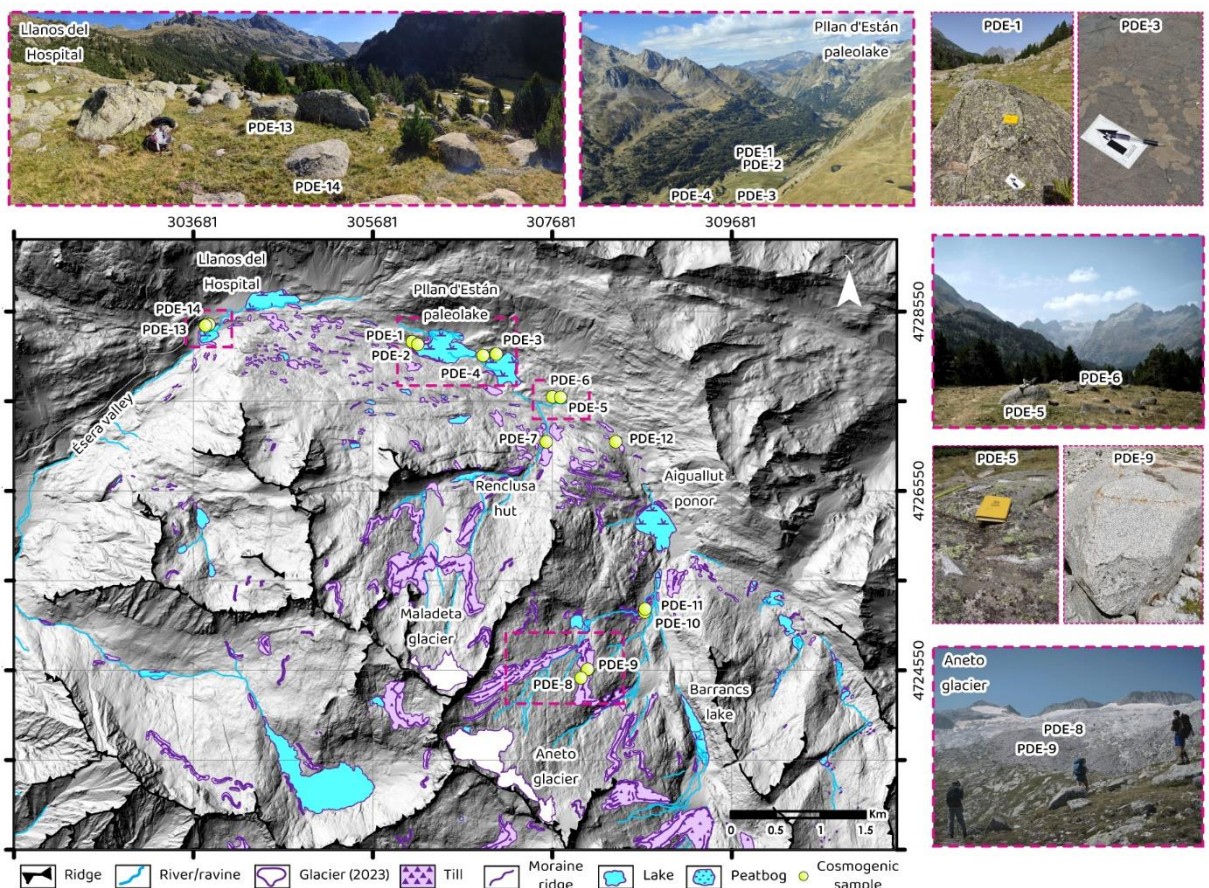

**Figure 2: Location of the cosmogenic samples and photographs of the boulders and the landscape for the geomorphological context. In the map, the yellow dots represent the location of each sample, and the pink squares the landscapes showed in each photo. Four samples (PDE-1, -2, -5 and -9) are showed in detail. The simplified geomorphological map was obtained and modified from Vidaller et al. (2024a). Hillshade obtained from Spanish National Geographic Institute.**

### 3.2 PaleoELAs

The reconstruction of paleoELAs of different glacial stages during the LGC represents a valuable tool for calculating temperature variations (Pellitero et al., 2019). Although there are numerous methods for determining the ELA, this study employs the area-altitude balance ratio (AABR) developed by Osmaston (2005), which is widely regarded as the most accurate (Serrano et al., 2004; Pellitero et al., 2015). To determine the paleoELAs using this method, the ELA Calculation Tool for





ArcGIS (Pellitero et al., 2015) was employed, with a balance ratio (BR, defined as the ratio between the accumulation area and the ablation area of a glacier) of 1.29 (Barr et al., 2022). For each calculated ELA, the glacier surface was reconstructed, taking into account glacier shapes (till, moraines, thresholds, ridges, cirques) and other geomorphological shapes (Vidaller et al., 2024a). The topographic reconstruction was conducted using a digital elevation model (DEM) with a resolution of 2 m/pixel, obtained from the *Centro Nacional de Información Geográfica* (CNIG). Given the favourable state of preservation of

the moraines from the Little Ice Age (LIA), the LIA ELAs obtained using the AABR method were compared with the elevation of the upper limit of the frontolateral moraines from this period. This upper limit is expected to correspond to the elevation at which the glacier began to melt, and thus the ELA elevation (also known as the MELA method). The 2023 ELA was obtained from orthophotos captured during a drone survey conducted in September 2023.

In order to ascertain the variation in temperature, it is necessary to consider the altitudinal gradient of temperature (AGT). The

AGT was calculated on the basis of climate data obtained from weather stations situated at Aneto peak (3404 m a.s.l.), Renclusa hut (2140 m a.s.l.), and Besurta hut (1929 m a.s.l.). The data for the years 2022 and 2020 were obtained from Posets-Maladeta Natural Park, AEMET and Clima y Nieve Pirineos database, respectively. Accordingly, the AGT in this valley is 0.525 °C/100 m (see *Supplementary Figure S2*), which is consistent with the AGT considered in the Pyrenees (García-Ruiz et al., 2024). Using this gradient and the varying paleoELA elevations of each phase with the 2023 ELA, the temperature variation can be

calculated using the following equation:

$$\Delta T = AGT \times (ELA_{2023} - ELA_{phase})$$

The reconstruction of paleotemperature has an estimated error of ±0.45°C (Barr et al., 2022), originated by the imprecision in the BR value.

## 4 Results

### 4.1 ¹⁰Be Exposure ages

The ¹⁰Be concentrations in the samples are presented in Table 1 for reference. The measured ¹⁰Be concentrations range from $0.13$ to $2.98 \times 10^5$ atoms/g(Qz), with typical errors of approximately 4%. However, some samples exhibit higher errors, reaching 8-13%, which can be attributed to two main factors: firstly, the limited availability of quartz mass (in PDE-3), and secondly, the relatively low ¹⁰Be concentrations observed in PDE-8 and PDE-9. The ²⁶Al concentration was determined for one bedrock

sample, PDE-3, and is $1.52 \times 10^6$ atoms/g(Qz), with an associated 20% error (see *Supplementary Table S2*). The ²⁶Al/¹⁰Be ratio for PDE-3 is calculated to be 8.0±1.7, which is indistinguishable from the nominal production rate ratio of 6.75 within the stated uncertainty (Balco et al., 2008). It can therefore be concluded that this bedrock surface has most likely experienced a simple exposure history, without any prolonged burial period (>2-300 ka).

The model minimum ¹⁰Be exposure ages, assuming no erosion and no snow shielding, range from 420 ± 36 years (PDE-8) and

390 ± 50 years (PDE-9) at the near Aneto peak to approximately 15 ka at the furthest point in the Ésera valley (PDE-13 and -14). It can be reasonably assumed that erosion rates of 3 mm ka⁻¹ would result in an increase in exposure ages of between 2





and 4% (see *Supplementary Table S3*). Furthermore, the application of snow shielding would also result in an increase in the calculated ages of between 0 and 6% for the majority of samples, with the exception of PDE-2 and -3, for which the offsets are larger (8% and 13%, respectively).




**Figure 3: Location of the cosmogenic samples (yellow dots) and ¹⁰Be exposure ages (pink squares) along the Ésera valley (geomorphological map modified from Vidaller et al., 2024a). Best ages (Table 1) are shown in ka. Hillshade obtained from a 5 m digital elevation model obtained from the Spanish National Geographic Institute.**




**Table 1. Results of cosmogenic $^{10}$Be measurements and exposure ages from Ésera valley. Uncertainties are in one sigma:**

| Field ID [a] | Latitude (ºN) | Long. (ºE) | Altitude (m) | $^{10}$Be [b] ($10^5$ atoms/g(Qz)) | Err [b] | Min. $^{10}$Be age (ka) [c] | Int. err | Ext. err | Best $^{10}$Be age (ka) [d] | Int. Err | Ext. err | Distance (km) [e] |
|---|---|---|---|---|---|---|---|---|---|---|---|---|
| PDE-1 | 42.6820 | 0.6337 | 1848 | 1.830 | 0.068 | 11.68 | 0.44 | 0.82 | 13.68 | 0.53 | 0.99 | 6.56 |
| PDE-2 | 42.6816 | 0.6344 | 1849 | 1.820 | 0.063 | 11.28 | 0.39 | 0.77 | 13.17 | 0.47 | 0.93 | 6.50 |
| PDE-3 | 42.6810 | 0.6452 | 1880 | 1.906 | 0.0015 | 11.34 | 0.92 | 1.14 | 13.24 | 1.11 | 1.37 | 5.65 |
| PDE-4 | 42.6807 | 0.6433 | 1854 | 2.213 | 0.077 | 13.87 | 0.49 | 0.95 | 14.34 | 0.52 | 1.02 | 5.77 |
| PDE-5 | 42.6766 | 0.6539 | 1933 | 2.206 | 0.080 | 12.47 | 0.45 | 0.87 | 13.12 | 0.49 | 0.94 | 4.79 |
| PDE-6 | 42.6768 | 0.6530 | 1937 | 2.136 | 0.074 | 11.96 | 0.42 | 0.82 | 12.54 | 0.45 | 0.89 | 4.87 |
| PDE-7 | 42.6723 | 0.6522 | 2039 | 2.474 | 0.085 | 13.07 | 0.45 | 0.90 | 13.55 | 0.49 | 0.96 | 2.68 |
| PDE-8 | 42.6487 | 0.6579 | 2660 | 1.399 | 0.012 | 0.420 | 0.036 | 0.044 | 0.429 | 0.037 | 0.045 | 0.11 |
| PDE-9 | 42.6496 | 0.6585 | 2634 | 1.299 | 0.017 | 0.390 | 0.050 | 0.055 | 0.401 | 0.051 | 0.057 | 0.11 |
| PDE-10 | 42.6555 | 0.6661 | 2189 | 2.155 | 0.001 | 10.65 | 0.51 | 0.81 | 11.07 | 0.55 | 0.87 | 2.05 |
| PDE-11 | 42.6555 | 0.6661 | 2195 | 2.209 | 0.080 | 10.56 | 0.38 | 0.73 | 10.81 | 0.40 | 0.77 | 2.05 |
| PDE-12 | 42.6726 | 0.6617 | 1995 | 2.204 | 0.079 | 12.32 | 0.44 | 0.85 | 13.33 | 0.50 | 0.96 | 3.94 |
| PDE-13 | 42.6831 | 0.6057 | 1755 | 1.955 | 0.069 | 15.03 | 0.53 | 1.04 | 16.64 | 0.62 | 1.20 | 8.94 |
| PDE-14 | 42.6830 | 0.6058 | 1744 | 2.048 | 0.081 | 15.23 | 0.61 | 1.09 | 16.26 | 0.67 | 1.21 | 8.95 |

**Internal (Int.) errors include only analytical errors and external (Ext.) errors include also systematic errors (such as errors associated with production rate and half-life). When compared to other geochronological data, external errors must be considered. (a) All sample and field details are summarized in *Supplementary Table S1*. (b) Details of laboratory data are described in text and**
**summarized in *Supplementary Table S2*. (c) Calculated assuming no erosion and no snow shielding. (d) Calculated assuming erosion rates at 3 mm/ka and snow shielding considering respective boulder heights, where for PDE1 and PDE2, which are located in the valley bottom, no boulder heights were considered (see text). (e) Distance between the sample location and the front of Aneto glacier in 2023 (except for PDE7, its distance relative to the Maladeta glacier in 2023).**





In general, the obtained ages were grouped into four clusters (Figure 4a), which corresponded to well-known periods during the last deglaciation (Clark et al., 2012; Oliva et al., 2019). The first pair of samples corresponded to the Oldest Dryas (OD)
(PDE-13 and -14), the second group were eight samples dated for the Bølling-Allerød period (B-A) (PDE-1 to -7 and -12), the third group belonged to the Early Holocene (PDE-10 and -11), and the last pair matched the LIA (PDE-8 and 9). The samples from the furthest downstream location (PDE-13 and -14), situated at Llanos del Hospital (Fig. 3), were retrieved from a moraine and represented the oldest ages in this study, with a date of 16-15 ka, which corresponds to the end of the OD. Eight samples situated between Pllan d'Están and Aiguallut (PDE1-7 and -12) exhibit highly similar ages, within the confines of their
respective error margins (Fig. 4a). These samples represent the deglaciation of the Ésera valley during the B-A period (14.6-12.9 ka). Samples PDE-10 and PDE-11 were obtained from a frontal moraine that corresponded to the transition to the Early Holocene. Finally, the samples collected from a moraine at the highest elevation in this study (PDE-8 and 9) indicated an age of approximately 0.4 ka, which suggests the extent of the LIA for the Aneto glacier (Fig. 3, Fig. 4 and Table 1).

The transition between the OD and the Allerød was marked by a retreat of the Ésera glacier of 2.4 km in ~2.7 ka, equal to a
deglaciation rate of 0.9 km ka$^{-1}$, indicating the onset of a progressively warmer period. Most of the samples from Pllan d'Están and Aiguallut were dated the final stage of the Allerød period, suggesting the fastest deglaciation of this valley, ~2.5 km in ~1 ka (Fig. 4b). Between the end of the Allerød period (~12.3 ka; PDE-12) and the Early Holocene (~10.6 ka; PDE-10 and -11), the glacier retreated slower, at a rate of ~1.0 km ka$^{-1}$ (retreated for 1.9 km in 1.7 ka), although the rate could have been even slower during the cold reversal corresponding to the Younger Dryas (YD, 12.9-11.7 ka; Palacios et al., 2016). Between the
Early Holocene and the LIA, the glacier retreated more slowly at 0.2 km ka$^{-1}$ (from positions of the samples PDE-10 and -11, to PDE-8 and 9; Fig. 4b).





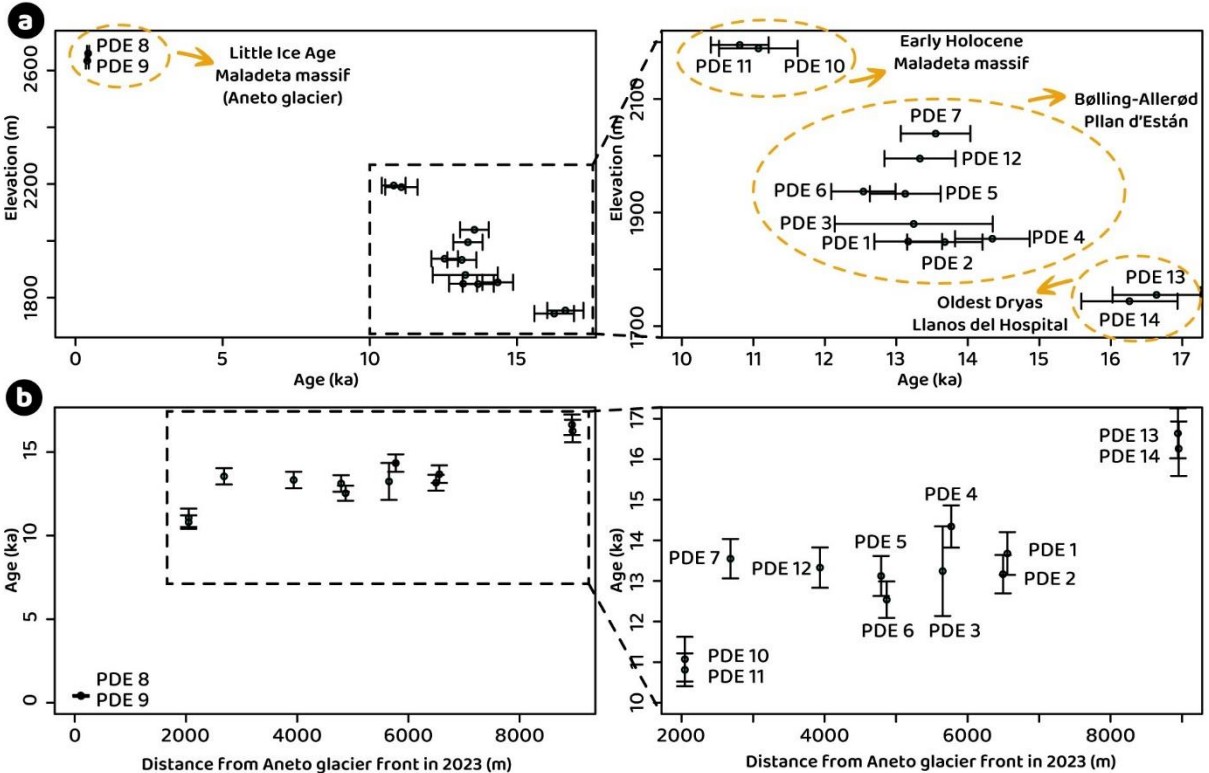

**Figure 4: (a) Distribution of dates according to the elevation of the samples. Clusters of ages are marked in yellow dash circles indicating the period associated to these samples. Right graph represents an enlargement of the left graph excluding LIA samples. (b) Distribution of dates according to the distance from the Aneto glacier front in 2023. Right graph represents an enlargement of the left graph excluding LIA samples, in order to focus on the retreatment and hold on of the deglaciation of the Ésera glacier.**

### 4.2 PaleoELAs

The reconstruction of the glacial phases during the deglaciation of the Ésera valley based on geomorphological evidence has enabled the definition of the paleoELAs and, to a reasonable approximation, the calculation of the temperature variation between those phases and the present day (2023). This has been achieved by following the equations indicated above. The results are presented in Table 2 for reference. In order to evaluate the methodology employed, a comparison was made between the ELA determined from drone survey images in 2023 and the theoretical ELA obtained with the AABR method. This comparison reveals a high degree of similarity between the two calculations, with an average difference of only 9 m. Furthermore, a similar exercise can be conducted using the LIA moraines, which are the most well-preserved moraines in the valley. The formation of these frontolateral moraines occurred at the onset of glacial retreat, coinciding with the initial stages of the ablation zone. Consequently, their elevations should align with the paleoELA of the LIA. In this instance, when solely considering the largest glaciers (Maladeta, Aneto and Tempestades), which exhibit the most well-preserved moraines and thus provide the most robust calculations, the discrepancy between the theoretical paleoELA of the LIA and the elevation of the moraine does not exceed 50 m, thereby validating our calculations.



The sedimentary sequence retrieved in Pllan d'Están (Fig. 1) establishes the position of the glacier front at 47 ka, coinciding
with the initial deposition of lacustrine sediments (Vidaller et al., 2024b). A comparison of the paleoELA for that time and the
current ELA allows the determination of a change of more than 3°C on average (Table 2). Similarly, during the deglaciation
process of the Ésera valley, other glacial phases and the associated paleoELAs and temperature changes are calculated. At 16
ka, the Ésera glacier reached Llanos del Hospital, as indicated by PDE-13 and -14 cosmogenic dates, representing a glacier
advance of 2 km down valley from the Pllan d'Están paleolake (Fig. 1). In accordance with the aforementioned evidence, the
recovered sediments from Pllan d'Están indicated a deposition in a subglacial environment (Vidaller et al., 2024a). The greatest
temperature increase occurred during the B-A period, with temperatures 0.4°C higher than at the start of the Early Holocene
period (Table 2). From the LIA to the present (year 2023), temperatures have risen by over 1°C (based on paleoELAs),
particularly in recent decades, leading to the rapid melting of the remaining glaciers.

| Age | 47 ka | 16 ka | 13.9 ka | 12.8 ka | 11 ka | 0.4 ka | Year 2023 |
|---|---|---|---|---|---|---|---|
| Period | MIS3 | Oldest Dryas | Onset Allerød | End Allerød | Early Holocene | Little Ice Age | Present |
| Phase | Pllan d'Están proglacial lake | Llanos del Hospital | Pllan d'Están subglacial lake | Aiguallut | Salterillo-Barrancs | Last advance | Very small glaciers |
| ELA (m a.s.l.) | 2517±42 | 2410±11 | 2519±47 | 2645±88 | 2778±77 | 2868±89 | 3099±140 |
| ΔT (ºC) Period-2023 | 3.1±0.2 | 3.6±0.1 | 3.0±0.2 | 2.4±0.5 | 1.7±0.4 | 1.1 ±0.5 | -- |

**Table 2: Weighted average paleoELA (with the AABR method and BR 1.29) of each phase and variation of temperature (increase
in all the cases) from each phase respect to year 2023. The uncertainty associate to the ELA corresponds with the standard deviation
of the ELA of the glaciers of each period. In the same way, the uncertainty of the variation of temperature is measured as the
variation of temperature associated to the standard deviation of the ELA for the different massifs and glaciers in the valley. These
data do not include the ±0.45ºC error associated to the imprecision in the BR value (Barr et al., 2022) which is constant along the
whole period. The phases mentioned in this table correspond with Figure 5 phases.**

## 5 Discussion

The combination of the new cosmogenic [10]Be exposure ages and the reconstruction of paleotemperatures presented in this
study, with the previous studies on the geomorphology of the valley (Vidaller et al., 2024a) and the Pllan d'Están sedimentary
sequence (Vidaller et al., 2024b) allows us to reconstruct the evolutionary history of the Ésera glacier with a great detail (Fig.
5). The almost absence of carbonates in the sedimentary record from Pllan d'Están together with a robust selection of dates
gives confidence to the obtained chronology based on [14]C dates, and complemented by OSL technique, resulting in similar
deglaciation ages than obtained from other lacustrine sequences located in the headwaters of different Pyrenean valleys (eg.
González-Sampériz et al., 2006). Therefore, this study based on new cosmogenic [10]Be exposure ages together with previous





data from Pllan d'Están paleolake allow to frame chronologically the different environments of the headwater of the Ésera

valley and the climatic implications.

## 5.1 Pre-16 ka phase

The oldest evidence of glacial evolution at the headwaters of the Ésera valley is located at the base of the Pllan d'Están paleolake sedimentary sequence characterized as till deposit. This till has been dated by OSL to 74.9 ka ± 7.3 (Vidaller et al., 2024b), as illustrated in Fig. 6a. This sediment is likely to represent the Pyrenean Last Glacial Maximum (PLGM) phase, as

described in previous studies for other Pyrenean valleys. These include dated moraines in the Aragón valley (68±7 ka; ~900 m a.s.l.; García-Ruiz et al., 2013), the Ara valley (55±11 ka; Sancho et al., 2018) and the Cinca valley (64±11 ka; 790 m a.s.l.); terraces associated with this glacial phase in the Gállego valley (66±4 ka and 74±10 ka; Peña et al., 2004) or a clastic deposit in the Granito cave with an age of 71.8±5.6 ka (Bartolomé et al., 2021). Therefore, the last glacial maximum extent in the Pyrenees does not correspond in time with the global Last Glacial Maximum (LGM), which is estimated to have occurred

between 23 and 18 ka, as observed in European glaciers for which the global LGM implied the greatest advance, erasing the glaciological record of previous periods (Cutler et al., 2003; Toucanne et al., 2023).

Following its maximum extent, the Ésera glacier is thought to have undergone advances and retreats associated with the abrupt temperature oscillations that characterised the MIS 3 (60-27 ka; Dansgaard et al., 1993). However, there is no information regarding glacial activity in the valley until 47.6 ka, when lacustrine sedimentation commenced in Pllan d'Están (Vidaller et

al., 2024b). This age correlates with the onset of one of the most significant warm interstadials during the MIS 3, the Greenland Interstadial (GI) 12 (Rasmussen et al., 2014). This suggests that the inception of the major deglaciation phase in the Ésera valley may coincide with the GI-12. At that time, Pllan d'Están was a proglacial bedrock-dammed lake (Figs. 5b, 6b), a type of proglacial lake common in large valleys (Otto, 2019). The glacier front was situated at an elevation of 1840 m a.s.l. In general, sedimentation in proglacial lakes is primarily influenced by the proximity to the glacier, resulting in the formation of

varves and rhythmites (Carrivick and Tweed, 2013). In Pllan d'Están, the initial lacustrine sediments were laminated and contained some carbonate eroded from the limestones in close proximity to the lake (Vidaller et al., 2024b). A similar pattern has been identified in other high-altitude lakes, including Marboré lake in the Central Pyrenees (Leunda et al., 2017) and Enol lake in the Cantabrian mountain range (Moreno et al., 2010). In these cases, the carbonate in the sediments has been preserved during relatively warm periods, when the dissolution of carbonates was prevented. It is noteworthy that the paleoELA at this

age was estimated to be 2517 m a.s.l. (Table 2), representing a temperature offset of 3.1 °C colder than the present. This temperature variation is consistent with marine sediment cores from the eastern margin of the Iberian Peninsula and the Western Mediterranean, which also indicate that GI-12 was the warmest interstadial (although colder than the present) within MIS 3 (Cacho et al., 1999; Martrat et al., 2007).

The carbonate content in Pllan d'Están sediments exhibited a precipitous decline following 34.8 ka BP marking a colder period.

The concomitant shifts in sedimentary facies and the low sedimentation rate are indicative of a subglacial lake environment (Vidaller et al., 2024b). This interpretation of very cold conditions is corroborated by palynological findings in other lacustrine



sequences, including El Portalet (González-Sampériz et al., 2006) and El Cañizar de Villarquemado (González-Sampériz et al., 2020) for the same time period. Furthermore, the existence of moraines in the Gállego valley at an elevation of 830 m a.s.l., dated using OSL to 36 ka (Lewis et al., 2009), also indicates a period of low temperatures and a probable glacial expansion.

The global LGM is linked to the most significant expansion of glaciers across northwest Europe, where glaciers reached their maximum extent during the Late Würmian period (MIS 2: 35-14 ka; Florineth and Schlüchter, 2000; Ivy-Ochs et al., 2008; Buoncristiani and Campy, 2011). While not ubiquitous across Europe, the advance was consistent in its magnitude. For instance, in the Maritime Alps, the estimated ELAs are as much as 450 m higher than those observed in the northern Apennines or the Corsican mountains. This suggests that the Italian side of the Maritime Alps experienced a relatively arid climate during
the LGM (Rettig et al., 2024). In spite the general importance of the LGM period in many European glaciers, this time period presents a more complex picture in the Pyrenees. In the headwaters of the Ésera valley, for instance, there are no published dates that correspond to this period (23-18 ka). In the Eastern Pyrenees, however, some LGM cosmogenic exposure ages from moraines at an elevation of 2183 m a.s.l. and evidence of the cold stage from lacustrine deposits have been reported. The existence of such sequences has been documented in the literature (Pallàs et al., 2006; Delmas et al., 2008, 2022; Rodes et al.,
2008; Delmas, 2015). The most supported explanation for this difference in the LGM glacier expression from west to east in the Pyrenees is related to the available humidity. The distribution of temperature and precipitation is mainly controlled by the elevation and the proximity to the Atlantic ocean or the Mediterranean sea (Cuadrat et al., 2007; García-Ruiz et al., 2015). The transition between the two climates is approximately in the Central Pyrenees, around the Maladeta massif.



**Figure 5: Evolution of the Ésera glacier since the PLGM. a) Illustrates the cold phase of maximum extent of the Ésera valley 75 ka ago. b) Shows a warmer phase during the MIS3 when Pllan d'Están paleolake was uncovered by ice for a time. c) Represents a new cold period and an advance of the Ésera glacier, whose front was located at Llanos del Hospital during the OD. d) At the onset of the Allerød a very fast deglaciation started. e) At the end of the Allerød, the Ésera valley had split in several cirques' glaciers. f) Shows the area covered by ice at the Early Holocene. g) Illustrates the extension of the glaciers of the north face of the Maladeta massif during the LIA. h) Represents the current situation where the glaciers are very small and withdrawn to the cirque's walls.**

## 5.2 Phase Llanos del Hospital (16 ka)

A significant indicator of the evolution of the Ésera glacier is the moraine dated at 16 ka in Llanos del Hospital at an elevation of 1750 m a.s.l. (PDE-13 and -14; Table 1 and Fig. 5). These ages coincide with a well-documented glacial advance in Europe,





the Older Dryas (OD) (approximately 17.5-15 ka), when Alpine glacier fronts reached 1400 m a.s.l. (Ivy-Ochs et al., 2006;

Kerschner and Ivy-Ochs, 2008; Darnault et al., 2012). The OD in the Pyrenees has been dated using moraines from the Eastern
Pyrenean valleys, in particular the Carlit massif (15 ka; Delmas et al., 2008), the Màniga-La Feixa platform (15.5 ka; Pallàs et
al., 2010), the Arriège valley (Delmas, 2015), the Noguera-Ribagorzana valley (Pallàs et al., 2006) and the Caldarés valley
(dates from 17.6 ka to 15.9 ka at an elevation of 2200-1400 m a.s.l.; Palacios et al., 2017b) have also yielded similar results.
Furthermore, the OD is well characterised in Pyrenean lakes and speleothems, indicating a sharp decrease in temperature

(Bernal-Wormull et al., 2021) and an increase in aridity (González-Sampériz et al., 2006; Morellón et al., 2009).

Reixach (2022) measured $^{10}$Be exposure ages from nearby sites at Llanos del Hospital. Their ages (10-18 ka) are scattered and
have relatively large errors (6-14%), and therefore the timing of glacial advance and retreat was not well constrained. Our new
exposure ages (16.6 ± 0.6 ka and 16.3 ± 0.7 ka; PDE-13 and -14 in Table 1), which are concordant between the two samples
and more precise, largely agree with the previous data when their ages are recalculated using the same parameters we used in

this study (e.g. $^{10}$Be production rate, geographical scaling scheme, snow density, post-depositional denudation rate).

The continuous sedimentary lacustrine sequence of Pllan d'Están and the location of the 16 ka moraine at a lower elevation in
the valley suggest a subglacial lake environment between 34-13 ka. Although it is very difficult to distinguish between
proglacial and subglacial sediments, the sediments of Pllan d'Están have different facies (Fig. 6), which help to distinguish the
two subenvironments. Proglacial sediments are usually characterised by the alternation of clay (summer months) and silt

(winter months) laminations forming varves (e.g. Smith and Ashley, 1985; Ringberg and Erlström, 1999; Palmer et al., 2008).
The rhythmic nature of the lamination is also marked by different grain sizes, indicating strong seasonality in a glacial
environment (Leonard and Reasoner, 1999; Ohlendorf et al., 2003). The coarse, angular silt laminae are deposited during the
melting season, whereas the fine silt laminae are deposited during the ice-covered season when fine particles settle by
suspension (Carrivick and Tweed, 2013). In subglacial lakes, sediments are usually homogenised towards the top of the

sequence, with sand bands and laminae (Livingstone et al., 2015), as occurred in the Pllan d'Están sedimentary sequence after
34 ka BP (Vidaller et al., 2024b). Furthermore, the sedimentation rate is lower than in the other lacustrine sequences of the
Pyrenees, such as El Portalet (González-Sampériz et al., 2006) or the Tramacastilla lakes (Jalut et al., 1982; Montserrat-Martí,
1992; García-Ruiz et al., 2003), which supports the subglacial lake condition at Pllan d'Están during ~34-13 ka BP.

The paleoELA at 16 ka was at 2410 m a.s.l., which implies that the average temperature was 3.6°C lower than the present.

375  Also, the paleoELA calculated is consistent with similar studied in the western Alps, where the paleoELA was at 2428 m a.s.l.
(Serra et al., 2022). Similarly, studies of marine sediments from the Iberian margin and the Western Mediterranean indicate a
comparable variation in temperature since the Heinrich Event 1 (Cacho et al., 2001; Martrat et al., 2007). Additionally, other
terrestrial records from the Iberian Peninsula indicate the occurrence of a particularly cold period at 16 ka. However, it should
be noted that quantitative temperature estimations are not yet available (Bernal-Wormull et al., 2021; Pérez-Mejías et al.,

380  2021).



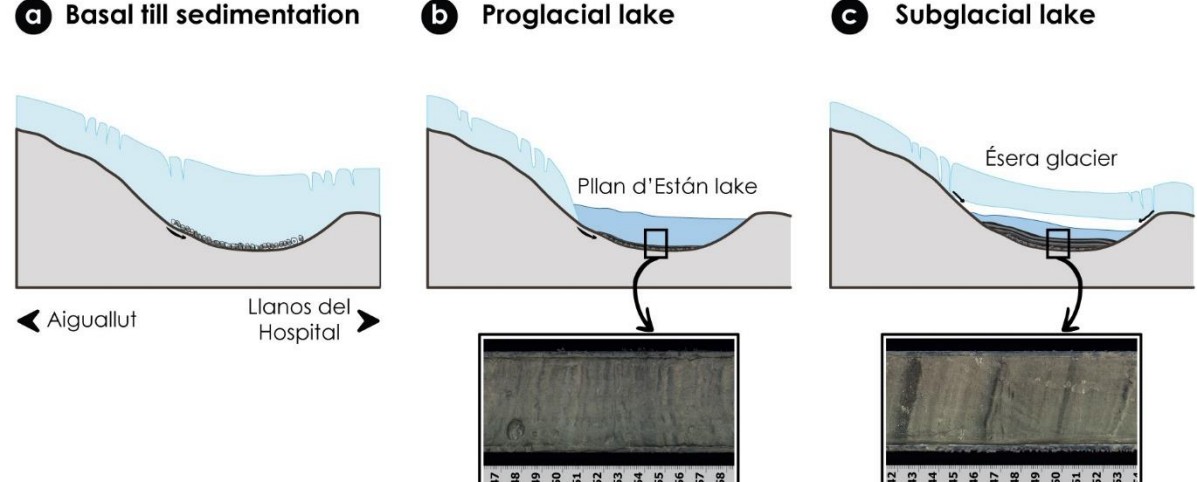

**Figure 6: Changes in sedimentation mechanism in Pllan d'Están overdeepening basin. a) Represents the first phase, dated at 75 ka when the glacier was excavating the basin and some basal till was deposited. b) Shows the sedimentation in Pllan d'Están paleolake when it was a proglacial lake during the relatively warm period of the MIS 3 (~47 ka). The mm-thick lamination suggests the sedimentation under proglacial lake condition. c) Represents the sedimentation in Pllan d'Están as a subglacial lake under the Ésera glacier. Banded sediments are associated to subglacial lakes. Both images of the sediment are obtained from the Pllan d'Están core, from different sections (Vidaller et al., 2024b).**

**5.3 Phase Pllan d'Están (13.7-12.8 ka)**

The majority of the cosmogenic samples included in this study were situated between Pllan d'Están basin and Aiguallut ponor (PDE1-7 and -12), exhibiting comparable ages within the confines of their respective error margins, which correspond to the B-A period (14.6-12.9 ka) (Fig. 5). This period was characterised by an increase in temperature and a change towards more humid conditions, as evidenced by both terrestrial (González-Sampériz et al., 2006; Bernal-Wormull et al., 2021; Vidaller et al., 2024b) and marine records (Fletcher et al., 2010). The cosmogenic dates indicate that the Ésera glacier commenced its retreat following the OD, approximately 16 ka. Therefore, during the Allerød period, the glacier front retreated from 1750 m a.s.l. (Llanos del Hospital) to 1995 m a.s.l. (Pllan d'Están), representing a rapid retreat of almost 4 km (at a rate of 2.3 km ka$^{-1}$). The accelerated deglaciation during the Allerød oscillation, in comparison to the Bølling, lends support to the hypothesis of a gradual increase in temperature, which contrasts with the abrupt change recorded in Europe at the onset of the Bølling (Vidaller et al., 2024b).

Similarly, other valleys in the Pyrenees exhibited comparable rapid retreat during the B-A, including the Bacivèr cirque (Oliva et al., 2021) and the Gállego valley. In the latter, the glacial front retreated from 1500 m a.s.l. to 2200 m a.s.l. between 14.6-11.7 ka (Palacios et al., 2015a). Similarly, glaciers situated in the Eastern Pyrenees (Delmas et al., 2008, 2011, 2023; Oliva et al., 2021) and the Ariège valley (Jomelli et al., 2020) retreated back to cirques during the B-A period. The shrinkage of the glaciers resulted in the formation of rock glaciers with debris covering glaciers below an elevation of 2800 m a.s.l. in numerous areas within the Pyrenees (Palacios et al., 2017a; Andrés et al., 2018; Oliva et al., 2021). Therefore, the B-A period is characterised by a rapid process of deglaciation, whereby glaciers in valley bottoms undergo a transformation into cirque



glaciers. In the Ésera valley, this phenomenon is exemplified by the Maladeta-Alba, Aneto-Tempestades and Escaletes glaciers (Fig. 5d and 5e).

In terms of temperature, at the conclusion of the Allerød, the paleoELA in the headwaters of the Ésera valley was recorded at 2645 m a.s.l., exhibiting a 1.3°C increase compared to the previous OD. This value is considerably lower than the estimated increase of 3-5°C derived from pollen records across Europe (Clark et al., 2012) or in marine cores surrounding the Iberian Peninsula (an increase of 5°C from Heinrich Event 1 to the Allerød period; Cacho et al., 2001; Martrat et al., 2007).

## 5.4 Phase Salterillo-Barrancs (11 ka)

The $^{10}$Be ages from PDE-10 and -11 (11.1±0.6 ka and 10.8±0.4 ka, respectively; Table 1) indicate that the Ésera glacier front was situated at 2046 m a.s.l. during the Early Holocene. These ages are somewhat older than the few preceding cosmogenic dates from analogous locations (~4-8 ka with one outlier at ~16 ka; Reixach, 2022). Similar ages have been reported from other regions within the Pyrenees. For example, in the La Cerdanya mountains (Southeastern Pyrenees), a rock glacier boulder was dated to 10.0±0.4 ka ($^{36}$Cl exposure ages) at an elevation of 2490 m a.s.l. (Palacios et al., 2015b). Similarly, in the Noguera-Ribagorzana valley at an elevation of 1721 m a.s.l., some cosmogenic ages have been reported, with the most recent being 10.4 ka (Pallàs et al., 2006). In the Carlit massif (Delmas et al., 2008), dates between 11-10.7 ka were obtained at an elevation of 2170-2180 m a.s.l. Thus, some polished bedrocks and boulders in the Gállego valley (Central Pyrenees) (Palacios et al., 2017a) were reported at an elevation of 2549-2719 m a.s.l. and dated with $^{10}$Be and $^{36}$Cl exposure ages, resulting in a date of 10.6±1.3 ka. The discrepancy in altitude between these samples may be attributed to their location in a western position, which is subject to a greater influence from an Atlantic climate.

In terms of temperature, the paleoELA of this phase indicates a difference of 1.7°C relative to the present (Table 2), and 0.7°C higher than the end of the B-A period. At a global scale, an increase of 1.3°C was recorded since the Early Holocene to current conditions (Shakun et al., 2012; Snyder, 2016). In the Eastern Swiss Alps, paleoELAs calculation showed an increase of 1.8°C (Joerin et al., 2008). This value contrasts again with the temperature change defined between the B-A and the onset of the Holocene in the Alboran sea, a change in SST of ~3°C (Cacho et al., 2001). The one study based on chironomids in the Pyrenees reveals a temperature difference of 1°C between the Early Holocene and the present (Tarrats et al., 2018), more coherent with our estimations. It is worth noting that many of such studies do not reconstruct annual record but are biased towards summer or spring temperature with biological indicators, and therefore care must be taken when so different studies are compared. Other studies based on chironomids too, but in the Australian and Swiss Alps, calculate an increase of 4°C since the Early Holocene to nowadays (mean July temperature; Ilyashuk et al., 2011) in the Austrian case and 1.7°C in the swiss case (Samartin et al., 2012).

## 5.5 Phase Little Ice Age (0.4 ka): the last advance of the glaciers

The LIA represents the final cold period documented in numerous mountainous regions across the globe, occurring during the 14[th] and 19[th] centuries (Grove, 2004; Solomina et al., 2016; Oliva et al., 2018a; García-Ruiz et al., 2020). This cold pulse was





characterised by low summer temperatures and snowy winters (Matthews and Briffa, 2005), implying a global cooling of 1-2°C lower than the present (Dyurgerov and Meier, 2000; Grove, 2004), with a likely larger temperature change in mountainous regions such as the Pyrenees. Some studies based on paleoELAs have determined that temperatures have increased by 2.5°C in the Eriste massif (Central Pyrenees) (Vidaller, 2018), 2°C in the Tendeñera and Sabocos mountain range (López-Moreno, 2000) and 0.9°C for the Posets massif (Serrano and Martín-Moreno, 2018), which is very similar to our own calculation of +1.1°C obtained from the Maladeta massif (Table 2). As indicated by data from the Spanish National Meteorology Agency (AEMET) and the Catalonia Meteorological Service (SMC), the temperature increase since the conclusion of the LIA in the Pyrenees region has been 1.1°C (Pérez-Zanón et al., 2017).

Since the LIA maximum, dated in this study with approx. 400 years, temperatures have increased almost continuously, and the glaciers have shrunk rapidly globally (Zemp et al., 2015; Oliva et al., 2018b). During the LIA there were 52 glaciers in the Pyrenees, that covered 2060 ha of glacier surface (Rico et al., 2017). Considering the case of Aneto glacier, the largest glacier in the Pyrenees, it has lost 64.7% of its area in the last 41 years (period 1981-2022), and its ice thickness has decreased by an average of 30.5 m (Vidaller et al., 2023). In addition, the occurrence of extremely hot and dry years, such as those observed in 2022 and 2023 summers, has accelerated the melting processes leading to a drastic degradation of the glacier and posing a high risk for its survival (Vidaller et al., 2023). In fact, at the current ELA, located at 3098.6 m a.s.l., very close to the walls of the cirque, snow/ice is absent in some areas indicating the demise of current glaciers and their accelerated melting.

## 6 Conclusions

The deglaciation of the Ésera valley since the PLGM was a complex process, comprising both advances and rapid retreats. The chronological sequence of deglaciation was determined through the analysis of dates of cosmogenic [10]Be from granitic boulders and polished bedrock along the headwaters of the Ésera valley. The excellent reproducibility of our data at each site, for example at Pllan d'Están (PDE-1, -2), in the vicinity of the Besurta hut (PDE-5, -6), at the LIA moraine (PDE-8, -9), the upper slope of Aiguallut (PDE-10, -11), or Llanos de Hospital (PDE-13, -14), makes a robust chronology of glacial phases in the Ésera valley. Furthermore, paleoELA calculations were conducted for various phases in order to estimate associated temperature changes.

Following the maximum extent of the Ésera glacier during the PLGM at approximately 75 ka, advances and retreats associated with climate oscillations during MIS 3 are likely to have occurred (60-27 ka). At the beginning of GI-12, a significant deglaciation occurred in the Ésera valley, marked by the formation of a proglacial lake in Pllan d'Están at approximately 47 ka. A subsequent glacial advance resulted in the lake being covered by the glacier once more time, and Pllan d'Están became a subglacial lake for an extended time period while sedimentation continued through the input of water and sediment via glacier crevasses. This subglacial lake condition was identified for the first time in the Pyrenees by combining the new [10]Be exposure ages presented in this study with those from previous sedimentological studies of the paleolake.




An important phase of the evolution of the Ésera glacier is marked by a moraine dated at 16 ka in Llanos del Hospital, which
represents the OD advance. This occurred when the temperature was 3.6°C colder than the present day. During the B-A
interstadial, the climate exhibited a large warming trend, particularly rapid during the Allerød. Our ELA analysis indicates an
increase of approximately 0.8°C in temperature (from the onset to the end of the Allerød) over a period of approximately 1,000
years, accompanied by a rapid retreat of the Ésera glacier from Pllan d'Están to near Aiguallut ponor (2.5 km). This study
offers a contrasting perspective to the majority of European paleoclimate records, which indicate an abrupt transition at the
onset of the Bølling period. The Early Holocene was characterised by an increase in temperature and the transformation of the
Ésera glacier from a valley glacier into smaller cirque glaciers. The LIA (0.4 ka) was the final cold period recorded in these
mountains and resulted in the formation of the largest and most well-preserved moraines in the valley.

## Team list

ASTER Team: G. Aumaître, K. Keddadouche, F. Zaïdi. CNRS, Aix Marseille Univ, IRD, INRAE, CEREGE, Aix-en-
Provence, France

## Author contribution

Ixeia Vidaller: Conceptualization, Formal Analysis, Funding acquisition, Investigation, Methodology, Validation,
Visualization, Writing original draft.
Toshiyuki Fujioka: Formal Analysis, Investigation, Methodology, Resources, Validation, Writing review and edition.
ASTER Team: Methodology, Resources.
Juan Ignacio López-Moreno: Conceptualization, Funding acquisition, Investigation, Methodology, Resources, Supervision,
Writing review and edition.
Ana Moreno: Conceptualization, Formal Analysis, Funding acquisition, Investigation, Methodology, Project administration,
Resources, Supervision, Validation, Visualization, Writing review and edition.

## Competing interests

The authors declare that they have no conflict of interest.

## Acknowledgement

This study was supported by the Spanish Autonomous Organism of National Parks project 2552/2020 (ORCHESTRA), the
Spanish project PID2019-106050RB-I00 (PYCACHU) and the Spanish project *Diputación Provincial de Huesca* Felix de



Azara 2021. Ixeia Vidaller is supported by the grant FPU18/04978 and is enrolled in the PhD programme at the University of Zaragoza.

We thank to Marcel Galofré for their help during the field work. Also, we thank Leticia Miguens, Angelli Pérez, Fernando Jiménez and Altug Hasözbek for their laboratory work in sample preparation for cosmogenic nuclide analysis and ICP-OES
measurements. ASTER AMS, national facility (CEREGE, Aix en Provence), is supported by the INSU/CNRS and IRD and member of AIX MARSEILLE PLATFORMS and REGEF networks. Also, we thank Ramón Pellitero for their help with the ELA calculation tool. Additionally, we would like to express our appreciation to AEMET for providing climatic data from the Renclusa and Besurta stations, to Marco from *Clima y Nieve Pirineo* for providing climatic data from Llanos del Hospital station, and to Posets-Maladeta Natural park for providing climatic data from Aneto station.

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
