# Peer review of "Geochronological reconstruction of the glacial evolution in the Ésera valley (Central Pyrenees) during the last deglaciation"

_Climate of the Past, 2024_

## Referee Comment (RC1)

**Geochronological reconstruction of the glacial evolution in the Ésera valley (Central Pyrenees) during ==the last deglaciation==**

*Please, use conventional stratigraphic terminology to better define the period cover by the study. "Last deglaciation" remains fuzzy.*

Ixeia Vidaller[1], Toshiyuki Fujioka[2], Juan Ignacio López-Moreno[1], Ana Moreno[1], ASTER Team[3,*]

[1] Instituto Pirenaico de Ecología, Consejo Superior de Investigaciones Científicas (IPE-CSIC), Zaragoza, Spain
5   [2] Centro Nacional de Investigación sobre la Evolución Humana (CENIEH), Burgos, Spain
[3] CNRS, Aix Marseille Univ, IRD, INRAE, CEREGE, Aix-en-Provence, France
[*] Aster Team: A full list of authors appears at the end of the paper.

*Correspondence to*: Ixeia Vidaller (ixeia@ipe.csic.es)

*Please, use conventional stratigraphic terminology, such as Late Pleistocene or other (MIS...) to better define the period cover by the study. "Last deglaciation" remain fuzzy.*

**Abstract.** The ==last deglaciation period== in the Pyrenees was distinguished by intricate glacier dynamics, encompassing a

10   multitude of advances and rapid glacier retreats ==that did not always align with the fluctuations observed in other European== *Caution with this kind of affirmation. This may be due to geochronological incertitudes related to available datings on ice margin deposits, climatic controls being homogenous at large (european) scale.* ==glaciers.== The Ésera valley, located in the Central Pyrenees (northern Spain), provides a distinctive opportunity to reconstruct *the Late Pleistocene* ==past climate== in high-mountain regions during the ==last deglaciation period.== Previous studies of glacial evolution in this area have *past climate and glacier fluctuations* employed a variety of methods, including the analysis of glacial lake sediments and detailed geomorphological studies of glacial landforms. This paper presents measurements of cosmogenic [10]Be exposure ages from glacial deposits and a polished

[revised manuscript text omitted]

*You should describe more accuracte the geomorphological markers used to reconstruct ELA. Each ELA must correspond to a specific frontal and or lateral moraine ridge because this kind of deposit (and landform), and only this one, is able to delineate the ice margin position of a specific glacial stade. You do that accurately for LIA but not for older glaciel stade. Please, do it for ALL glacial stades.*

175 ArcGIS (Pellitero et al., 2015) was employed, with a balance ratio (BR, defined as the ratio between the accumulation area and the ablation area of a glacier) of 1.29 (Barr et al., 2022). For each calculated ELA, the glacier surface was reconstructed, taking into account glacier shapes (till, moraines, thresholds, ridges, cirques) and other geomorphological shapes (Vidaller et al., 2024a). The topographic reconstruction was conducted using a digital elevation model (DEM) with a resolution of 2 m/pixel, obtained from the *Centro Nacional de Información Geográfica* (CNIG). Given the favourable state of preservation of

*Older moraines are less preserved but they exist. Hence, you have to tell us which moraines you used to reconstruct ELA older than LIA.*

180 the moraines from the Little Ice Age (LIA), the LIA ELAs obtained using the AABR method were compared with the elevation of the upper limit of the frontolateral moraines from this period. This upper limit is expected to correspond to the elevation at

*This is unclear... Do you talk about "lateral moraine method" after Porter (2001)?*

which the glacier began to melt, and thus the ELA elevation (also known as the MELA method). The 2023 ELA was obtained

*Thus, it is the "lateral moraine method" from porter (2001). Note this method give an under-estimation of the ELA and is very less accurate than the AABR method. I'm not sure it is really usefull here.*

from orthophotos captured during a drone survey conducted in September 2023.

*You do not explain how do you constrain the ELA after orthophotos??? To constrain current ELA, you need to record data from glaciological balises on the field.*

[revised manuscript text omitted]

*Ice extent at 16 ka is based on PDE13 and 14, i.e. Llanos de Hospital glacial stillstand stade. But you do not indicate on which samples (and thus on whch moraines ridges and glacial stades) you calculate ELA at 13.9, 12.8 and 11 ka. These informations are essential to guarantee the reproducibility of these results.*

**5 Discussion**

*Given the importance of data from Plan d'Estan (OSL and 14C datings from lacustrine sequence), I suggest you integrate to this paper a figure sumarizing data that are used here (in this paper) to reconstruct esera glacier fluctuations.*

The combination of the new cosmogenic [10]Be exposure ages and the reconstruction of paleotemperatures presented in this study, with the previous studies on the geomorphology of the valley (Vidaller et al., 2024a) and the Pllan d'Están sedimentary sequence (Vidaller et al., 2024b) allows us to reconstruct the evolutionary history of the Ésera glacier with a great detail (Fig.

285   5). The almost absence of carbonates in the sedimentary record from Pllan d'Están together with a robust selection of dates gives confidence to the obtained chronology based on [14]C dates, and complemented by OSL technique, resulting in similar deglaciation ages than obtained from other lacustrine sequences located in the headwaters of different Pyrenean valleys (eg.

*Please, do not be so allusive. Detail what you mean.*

González-Sampériz et al., 2006). Therefore, this study based on new cosmogenic [10]Be exposure ages together with previous

[Figure]

*Given the importance of data from Plan d'Estan (OSL and 14C datings from lacustrine sequence), I suggest you integrate to this paper a figure sumarizing data that are used here (in this paper) to reconstruct esera glacier fluctuations.*

data from Pllan d'Están paleolake allow to frame chronologically the different environments of the headwater of the Ésera

290    valley and the climatic implications.

**5.1 Pre-16 ka phase**

The oldest evidence of glacial evolution at the headwaters of the Ésera valley is located at the base of the Pllan d'Están paleolake sedimentary sequence characterized as till deposit. This till has been dated by OSL to 74.9 ka ± 7.3 (Vidaller et al., 2024b), as illustrated in Fig. 6a. This sediment is likely to represent the Pyrenean Last Glacial Maximum (PLGM) phase, as

295    described in previous studies for other Pyrenean valleys. These include dated moraines in the Aragón valley (68±7 ka; ~900 *This is a OSL dating from fluvioglacial deposit (terrace at + 20m)*

m a.s.l.; García-Ruiz et al., 2013), the Ara valley (55±11 ka; Sancho et al., 2018) and the Cinca valley (64±11 ka; 790 m a.s.l.); *The error bar for this age is 4.5 ka and not 11 ka.* *This age from 3 OSL datings on a fluvioglacial deposit (terrace Qt7)*

terraces associated with this glacial phase in the Gállego valley (66±4 ka and 74±10 ka; Peña et al., 2004) or a clastic deposit *In Pyrenees, they are other evidences of glacial advances at the begining of the Late Pleistocene : see Pallas et al., 2010 and Delmas et al., 2011.*

[revised manuscript text omitted]

Hence you must assume that plan Estan was covered by ice at the time of Global LGM and at the time of Oldest Dryas.

the valley suggest a subglacial lake environment between 34-13 ka. Although it is very difficult to distinguish between

In order to well understand this sentence, it is very important to give (in this paper) a figure reportig all 14C and OSL datings (with error bar) obtained in the lacustrine sequence of plan d'Estan.

proglacial and subglacial sediments, the sediments of Pllan d'Están have different facies (Fig. 6), which help to distinguish the
two subenvironments. Proglacial sediments are usually characterised by the alternation of clay (summer months) and silt
365 (winter months) laminations forming varves (e.g. Smith and Ashley, 1985; Ringberg and Erlström, 1999; Palmer et al., 2008).
The rhythmic nature of the lamination is also marked by different grain sizes, indicating strong seasonality in a glacial
environment (Leonard and Reasoner, 1999; Ohlendorf et al., 2003). The coarse, angular silt laminae are deposited during the
melting season, whereas the fine silt laminae are deposited during the ice-covered season when fine particles settle by
suspension (Carrivick and Tweed, 2013). In subglacial lakes, sediments are usually homogenised towards the top of the
370 sequence, with sand bands and laminae (Livingstone et al., 2015), as occurred in the Pllan d'Están sedimentary sequence after
34 ka BP (Vidaller et al., 2024b). Furthermore, the sedimentation rate is lower than in the other lacustrine sequences of the
Pyrenees, such as El Portalet (González-Sampériz et al., 2006) or the Tramacastilla lakes (Jalut et al., 1982; Montserrat-Martí,
1992; García-Ruiz et al., 2003), which supports the subglacial lake condition at Pllan d'Están during ~34-13 ka BP.

Hence you must assume that plan Estan was covered by ice at the time of Global LGM and at the time of Oldest Dryas.

[revised manuscript text omitted]

---

## Author Comment (AC1)

**Reviewer 2**

Review of the manuscript cp-2024-75 "Geochronological reconstruction of the glacial evolution in the Ésera valley (Central Pyrenees) during the last deglaciation" by Vidaller et al.

This manuscript deals with the reconstruction of different stages of glacier retreat and readvances (here called last deglaciation) in northern Spain. The authors use cosmo nuclide dating, as well as isotope ratios to constrain the glaciation history. The methods are robust, even though paleoELA is subject to quite some uncertainties in terms of interpretation of impacts of temperature versus precipitation changes. It would be good to defend this point in a more convincing manner — with this being said, it is a standard practice in the geomorphological community and is thus acceptable. One very interesting discovery is the subglacial paleolake that existed under the glacier during the glacier readvance prior to its final, relatively smooth retreat.

I enjoyed reading the manuscript – it is relatively well written, presents a nice story and new, previously non-existent data that are high quality and well-aligned in chronology as opposed to earlier studies in the same area characterized by a large scattering. I particularly liked the discussion in light of existing proxy data in the region and globally. Nicely done. I recommend publication after minor revisions that are listed below.

We greatly appreciate this constructive remark and concur on the significance of examining lacustrine records linked to glacier evolution, as they provide crucial evidence for understanding the timing and patterns of glacial retreat. The Pllan d'Están sequence constitutes an exemplary case study illustrating the potential of this research approach.

**General comments**

Please, edit for typos. There are quite many of them. I will pass the file with some typos marked through the editorial system.

Answer (hereinafter A): Thank you, we have reviewed all the typos.

The title with "last deglaciation" is a bit misleading because the authors also discuss the PLGM conditions at 75 ka and MIS3 evidence of glacier shrinking. I understand that it could be inferred that the last deglaciation actually started at 75 ka and continued until now but this is not commonplace to state something like this and besides the choice of a journal (CP, which is focusing on the climate reconstructions, not on paleoglaciology or glacial geomorphology as such) requires that the authors make their messages very clear and use generic language. I therefore suggest that the authors simplify their narrative for the cross-discipline communities.

A: It is true that some terms are too specific. We have tried to simplify the nomenclature using more generic language, including in the title, to make it understandable for cross-disciplinary communities.

Again, for PaleoELA calculations – a very old publication. I understand that everyone is using ELA approach but the authors could at least add a discussion of how alternative methods could differ in terms of climate interpretation and how robust it is.

A: It is true that the references mentioned in the text are a bit old, but there are not any recent study defending trying new method, all the recent investigations use the same method (AABR), considering its limitations. In this sense, a new paragraph was added in order to clarify this issue: "There are several methods to determine the ELA for a glacier, but the most accurate method is the Area x Altitude Balance Ratio (AABR; Benn et al., 2005), although this method presents challenges associated with the reconstruction of past glacier extensions, particularly the identification and dating of geomorphological features that are not always well preserved to reconstruct the surface of glaciers, overthought this approach remains a valuable method for understanding past thermal changes in mountainous environments (Pellitero et al., 2019)."

The glacier has been retreating since the LIA. Why didn't you use more robust, recent observations to validate your ELA calculations? There are plenty of remote-sensed studies, aerial surveys, and maybe even some repeat photography. Besides, there are climate reanalysis data and observations to validate your choices of parameters.

A: Here, in the Pyrenees, there are no remote-sense studies or aerial surveys for the LIA, only some pictures and old photos from the first alpinist. Even thought, there are very well conserved moraines for this cold period, so the best option is to reconstruct the glacier surface using the glacial geomorphology feature. A sentence was added: "Unfortunately, drawings or photographs of enough quality for the LIA period to better constrain the ELA are not available, so the best option was to use the glacial geomorphology feature to reconstruct the glacier surface."

In the line 255 you mention equations. What equations are we talking about? There are no eq. numbers, neither do I find many equations in this work. Namely it is just one it seems. Please, clarify.

A: This sentence has been changed as: "This has been achieved considering the ELA obtained with the AABR method for each moment and the AGT calculated with eq. 1"

Temperature lapse rate does not only depend on elevation but also on the season (can go from 4.5C/km in summer to 10C/km in winter) and even on the background climate. Please, add it in your discussion of limitations. I think such a section might need to be introduced in the appendix. Since this is part of a PhD study, such section is anyway needed in the thesis.

A: For this study mean annual temperature has been used, but as R1 the temperature lapse could vary between different periods. A new sentence has been added in the review manuscript to mark the limitations of the method: "PaleoELAs have been demonstrated to serve as effective proxies in the determination of temperature variations during periods for which instrumental records are unavailable. However, it should be noted that the method is not without its limitations. In certain instances, such as the present case, the temperature variation values obtained through the utilisation of paleoELAs do not correspond with those obtained through the application of alternative proxies."

**Minor comments:**

Line 260: Not obvious that the formation of moraines is marking the onset of glacier retreat. Aren't they accumulate during stillstands? Also, "ablation zone" seems out of context here. Further discussion/explanations are needed here.

A: This sentence has been removed.

Line 272: I don't understand these calculations. Please, explain in a better way.

A: To variation of temperature is based in the variation of the paleoELA. Considering a stable AGT throughout time, the difference of temperature is based on the difference of the elevation of the paleoELAs. A: This sentence has been changed as: "This has been achieved considering the ELA obtained with the AABR method for each moment and the AGT calculated with eq. 1"

Table 2: Correct "Period - 2023". It is the other way around. Clarify if it is a summer temperature difference or mean annual?

A: This sentence has been added to the table description: "This has been achieved considering the ELA obtained with the AABR method for each moment and the AGT calculated with eq. 1"

In section 5.1 you use a sediment core record dated by the OSL method. Was it not possible to cross-correlate temperature anomalies?

A: If we understand correctly, you ask to correlate temperature obtained with paleoELAs and with sediment data. In this case we did not inferred temperature data of the sediment core, only the type of ambient (e.g. a forest surrounding the lake or cold climate that avoid the development of organic matter correctly).

The statements about the early PLGM in section 5.1 are not inclusive. For example, the Barents-Kara Sea ice sheet reached its local LGM during MIS5. A lot of glaciers in Asia did too. The Patagonian ice sheet did so during MIS3, as well as glaciers in New Zealand.

A: It is right, this statement refers to a European context. This has been clarified in the reviewed text: "Therefore, the last glacial maximum extent in the Pyrenees does not correspond in time with the global Last Glacial Maximum (LGM) considering a European context, which is estimated to have occurred between 23 and 18 ka, as observed in European glaciers for which the global LGM implied the greatest advance, erasing the glaciological record of previous periods (Cutler et al., 2003; Toucanne et al., 2023)."

Line 325: Misleading sentence – In many parts of the world glaciers during the Penultimate LGM were more extensive. Indicate that this is only for the last glacial period.

**A: Change as suggested.**

Lines 356-357: Why is that? How do you justify that your choice of parameters is more reliable? Uncertainties in delta T estimates should be explicitly discussed.

A: Because our margin of error in dating is smaller and because in almost all locations we take samples from two nearby blocks and the results are very similar. In previous studies, not only is the margin of error in dating greater, but also, in some cases, the dates of nearby blocks are very different.

Line 403: I am not sure I understand this sentence. Rock or debris-covered glaciers or both?

A: Both, this has been clarified.

Lines 408 – 411: How do you explain such a disparity? What about the role of precipitation change?

A: As in a previous comment we have answered, the method of the proxy has limitations, this is discussed in lines 353-357 of the revised manuscript.

Line 420: The formulation is weird.

A: This sentence has been rewritten as: "Thus, some polished bedrocks at an elevation of 2549-2719 m a.s.l in the Gállego valley (Central Pyrenees), were dated with 10Be and 36Cl exposure ages, resulting in a date of 10.6±1.3 ka (Palacios et al., 2017a)."

Line 431: I agree regarding biological indicators but isn't it the same for ELA to some extent?

A: I agree with you, the ELA determines the elevation of the iso 0°C during the ablation period, so during summer. In this sense this discussion has been deleted.

Lines 439-440: Why so?

A: These values were determined in other studies.

Line 462: What mechanism can explain such an early local LGM?

A: In this sense, in the rest of Europe, there was also a glaciation around 70-60 ka, but there during the LGM (23-18 ka) glacier advanced more than during the 70-60 expansion. In the Pyrenees, during the LGM the weather was cold but arid, avoiding big accumulation of snow, and consequently the expansion of glaciers.

How reliable are reconstructions of glacier retreat followed by a readvance? What makes your methodology trustworthy? Explain for paleoclimatologists.

---

## Author Comment (AC2)

**Reviewer 1**

Comment on "Geochronological reconstruction of the glacial evolution in the Ésera valley (Central Pyrenees) during the last deglaciation" by Ixeia Vidaller, Toshiyuki Fujioka, Juan Ignacio López-Moreno, Ana Moreno, ASTER Team.

This paper presents a series of 14 surface exposure dating on boulders and polished bedrock in Ésera valley (Central Pyrenees, Spain). These new data are cross-checked with OSL and 14C dating previously published (Vidaller et al., 2024) and obtained from a sedimentary sequence located at Pllan d'Están, ~ 7 km downstream of the LIA moraines and the current glaciers of the Maladeta massif. The Pllan d'Están sequence records three successive units: (i) a sub-glacial till at the bottom of the core, (ii) a proglacial lacustrine deposit in the middle of the sequence, (iii) sub-glacial lacustrine deposit at the top of the sequence. Both series of data allow the Late Pleistocene fluctuations of the Ésera glacier to be reconstructed. Additionally, authors quantify changes in ELA position and in temperature anomalies (with respect to present) for 6 glacial stages respectively dated at 47 ka, 16 ka, 13.9 ka, 12.8 ka, 11 ka and 0.4 ka.

**General comments**

The most interesting (and innovative) result of this work is the evidence of a major deglaciation of the Ésera valley during MIS 3. Indeed, around 47 ka, the terminal position of the Ésera glacier was located upstream of the Plan d'Están, as indicated by an OSL dating at 46.7±2.9 ka within the proglacial unit that overlies the basal till. This result gives a substantial advance on Late Pleistocene glacier fluctuations in the Pyrenees because, until now, most of the data produced in this mountain were obtained from moraines ridges or paleolakes close to the PLGM and, therefore, they provided information on the maximum glacial advances and very scarcely on episodes of glacial retreat.

We really appreciate this positive comment and agree with Reviewer 1 about the importance of studying lake records associated to glacier evolution to get information on glacial retreat phases. Pllan d'Están sequence is an excellent example of this type of studies.

Beyond this major result, two other issues should be addressed more explicitly in this paper.

What was the extent of the Ésera glacier at the time of the global LGM? I aware that available
data do not allow the terminal position of the Ésera glacier to be located at the time of the
global LGM, but available data allow to consider if Ésera glacier covered (or not) Pllan d'Están
at the time of the global LGM.

Answer (hereinafter A): Certainly, the extend of the Ésera glacier is an issue of major interest that, unfortunately, it is difficult to answer properly with our available data. Up to now, no cosmogenic exposure dates in the valley correspond with the LGM but, thanks to the OD dated moraine (~16 ka) and the characterization of Pllan d'Están sediments, we have some clues about the extension of the glacier that are now better included in the manuscript. Thus, at 47 ka Pllan d'Están is defined as a proglacial lake, with laminated (almost varved) sediments. Later, at 34 ka BP, the sedimentation changes and the laminations become cm-thick bands and the carbonate amount decrease. This change is interpreted as the onset of lacustrine subglacial sedimentation,

according to some sedimentological references (eg. Livingstone et al., 2015). The next information following the temporal line is the OD-dated moraine at Llanos del Hospital, down valley from Pllan d'Están. Therefore, during the LGM, the available evidences indicate that Pllan d'Están was covered by ice; therefore, the glacier was located, at least, at that position. Some sentences will be added to the text to clarify this question.

A similar question arises in relation to the Younger Dryas. What was the extent of the Ésera glacier at the time of the Younger Dryas? Authors assume that the terminal position of the Ésera glacier was located upstream to Pllan d'Están because 14C dating around 13 ka were obtained at the top of the upper subglacial lacustrine unit. However, authors do not really correlate this evidence deduced from Pllan d'Están 14C dating with surfaces exposure dating on boulder located upstream Pllan d'Están filling and with moraines ridges located in the same place.

A: To discriminate the position of the glacier during the Younger Dryas (YD) we have again several arguments but none is very conclusive since we don't have any moraine dated covering that time period. The sediments in the Pllan d'Están sequence associated to the YD (well dated in this case by 14C) correspond with an interval of ca. 10 cm composed of banded, organic-rich silts deposited in a lacustrine environment (Facies F, with some flood layers – Facies C). Clearly, this does not correspond with a subglacial environment. Besides, several cosmogenic exposure dates associate to the Bølling-Allerød (B-A) period mark the position of the Ésera glacier upstream Pllan d'Están. We suspect that the glacier advance associated to the YD in this valley was a small one, probably the front would be located between the position of the B-A dates and Pllan d'Están. Thus, according to Rev1, we need to wait for other conclusive evidences to place the extension of the YD glacier in a more robust way but we interpret from the sediments that the glacier was not covering the lake

Overall, in order to enable readers to make up their own minds on the subject, it would be useful to provide in this paper a figure that summarizes information contained in the Pllan d'Están sequence that are essential for reasoning about the chronology of glacial fluctuations (thickness and content of each sedimentary unit, location of OSL and 14C dating within the sequence, all 14C with their error bar and not only those retains by the Bayesian model age...).

A: We agree with the reviewer about the convenience of including a figure summarizing Pllan d'Están sequence. The new version will include that figure, combining sedimentary characteristics, vegetation cover and chronological information. That figure allows correlating sedimentological information with the TCN samples presented in this manuscript.

Regarding 14C dates, we include only the ones considered to construct the chronology, since the other ones were discarded due to their location in core sections not used to build the composite sequence. The Bayesian age model includes all the valid ages (19 dates by 14C and 3 by OSL) and no reversal was found. In Fig. 3 of Vidaller et al., 2024, black dots indicated the dates that were discarded before constructing the chronology but they were not included in the model and not included in new Figure 5 either.

Moreover, it would be interesting to analyse the TCN results with respect to the location within the sequence of moraines ridges (hospital moraine, plan de llanos moraine, Aiguallut moraine, etc...) because these frontal and lateral moraines ridges delineate the ice extent of the Ésera glacier

for several glacial still stand stadial. TCN dating from boulders located on frontal and/or lateral moraines ridges allow these events (these glacial still stand stadial) to be dated.

A: In this sense several solutions have been proposed. Table 2 in the new document, has been completed with two new rows, one indicating the samples that define each phase, and the other one, indicates the type of deposit dated. On the other hand, in Figure 2 with the position of the TCN dating samples, a different colour is used to each type of sample (yellow for moraine ridge boulder, orange for till boulders, green for erratic boulders and purple for polished surface).

In the same way, you should describe more accurately the geomorphological markers used to reconstruct ELAs. Each ELA must correspond to a specific frontal and/or lateral moraine ridge because this kind of deposit (and landform), and only this one, is able to delineate the ice margin position of a specific glacial stage. You do that accurately for LIA but not for older glacial stage. Please, do it for ALL glacial stages.

A: Some sentences with respect to this issue has been added: "For each calculated ELA, the glacier surface was reconstructed, considering the frontal moraines and till previously mapped and dated with cosmogenic isotopes. To determine the lateral extension of the Ésera glacier for each extent lateral moraines, till and in some cases very clear thresholds that avoid the expansion of very thin layer of ice previously mapped glacier shapes (till, moraines, thresholds, ridges, cirques) and other geomorphological shapes (Vidaller et al., 2024a)."

Together, the two previous comments should help readers to better understand the chronological milestones at 13.9, 12.8 and 11 ka reported in Table 2. Indeed, because ice margin deposits used to reconstruct ice extent at several time of the Late Pleistocene is not well explained and because TCN dating are not clearly located with respect to moraines ridges, we miss information to understand how you produced milestones at 13.9, 12.8 and 11 ka associated to ELA reconstructions.

A: Two new rows has been added to the table to make clearer how we produced the ELA reconstructions for 13.9, 12.8 and 11 ka. Still, we agree with Rev1 that not all of our TCN samples were obtained from moraine ridges.

| Age                    | 47 ka                            | 16 ka                  | 13.9 ka                          | 12.8 ka      | 11 ka                   | 0.4 ka                | 2023                |
|------------------------|----------------------------------|------------------------|----------------------------------|--------------|-------------------------|-----------------------|---------------------|
| Sample                 | Pllan d'Están
core            | PDE-13
PDE-14       | PDE-1                            | PDE-12       | PDE-10
PDE-11        | PDE-8
PDE-9        | Drone images        |
| Sample type            | Core                             | Moraine ridge boulder  | Till boulder                     | Till boulder | Moraine ridge boulder   | Moraine ridge boulder | -                   |
| Period                 | MIS3                             | Oldest Dryas           | Onset Allerød                    | End Allerød  | Early Holocene          | Little Ice Age        | Present             |
| Phase                  | Pllan d'Están
proglacial lake | Llanos del
Hospital | Pllan d'Están
subglacial lake | Aiguallut    | Salterillo-
Barrancs | Last advance          | Very small glaciers |
| ELA (m a.s.l.)         | 2517±42                          | 2410±11                | 2519±47                          | 2645±88      | 2778±77                 | 2868±89               | 3099±140            |
| ΔT (°C)
Period-2023 | 3.1±0.2                          | 3.6±0.1                | 3.0±0.2                          | 2.4±0.5      | 1.7±0.4                 | 1.1 ±0.5              |                     |

Note that the concept of "deglaciation" used numerously in the manuscript (even in the title) is not enough clear in term of time. It does not allow to identify the period covers by the paper. I think you should use conventional stratigraphic terminology such as "Late Pleistocene".

A: We agree with the reviewer about the misleading term of "deglaciation". In some papers, it is understood as the period starting with the onset of the Bølling (14.6 ka BP) while in papers more related to glacier phases is much more ambiguous. Therefore, this term has been changed in all the manuscript by Late Pleistocene and also in some cases where there were references to the LIA, the term Holocene has been added too. The title is changed accordingly.

**Minor comments:**

Title: Please, use conventional stratigraphic terminology to better define the period cover by the study. "Last deglaciation" remains fuzzy.

A: The title of the manuscript has been changed to: "Geochronological reconstruction of the glacial evolution in the Ésera valley (Central Pyrenees) during the Late Pleistocene and Holocene".

Line 9: Please, use conventional stratigraphic terminology, such as Late Pleistocene or other (MIS...) to better define the period cover by the study. "Last deglaciation" remain fuzzy.

A: This sentence has been rewritten as: "Since the MIS (Marine Isotope Stage) 4 in the Pyrenees was...".

Line 10: "retreats that did not always align with the fluctuations observed in other European glaciers". Caution with this kind of affirmation. This may be due to geochronological uncertainty related to available dates on ice margin deposits, climatic controls being homogenous at large (european) scale.

A: Thank you, we agree with you, this is an open question. We have reformulated the sentence as: "Since the MIS (Marine Isotope Stage) 4 in the Pyrenees was distinguished by intricate glacier dynamics, that might encompass some glacial advances and retreats that did not always align with the fluctuations observed in other European glaciers."

Line 12: Add "and glacier fluctuations" after "past climate"

A: Change as suggested.

Line 12: Change "last deglaciation period" by "Late Pleistocene"

A: This sentence has been changed as: "...since the Pyrenean Last Glacial Maximum (PLGM)". In other similar sentences, last deglaciation period has been changed as Late Pleistocene and Holocene to be more accurate.

Line 16: Change "last deglaciation period" by "Late Pleistocene"

A: This sentence has been removed

Line 17: Add acronym PLGM.

**A: Change as suggested.**

Line 19: "a new glacial advance resulted forming the Llanos del Hospital moraine (~16 ka)" Readers wants to know what is your opinion about the extent of the Ésera glacier at the time of the global LGM (19-24/26 ka cal BP)

A: We have responded to this question above and that information has been added in the discussion: "In the headwaters of the Ésera valley, for instance, there are no published dates that correspond to this period (23-18 ka), but surely Pllan d'Están was covered by ice, still being a subglacial lake."

Line 29: "thickness and surface variations". Thickness, surface, and thus volume of current and formers glaciers also depends on catchment hypsometry (this parameter controls the volume of ice input each year in the accumulation area)

A: We agree with you; this aspect has been added: "Consequently, their thickness and surface variations, in spite they also depend on the catchment hypsometry, are regarded as one of the most informative proxies for climate change in mountainous regions".

Line 31: Change "evolution" by "fluctuation"

A: Change as suggested.

Line 36: Change "erratic boulders, moraines..." by "erratic boulders on moraines"

A: This sentence has been changed as: "Despite the discontinuous nature of glacial deposits, the dating of erratic boulders on moraine ridges, or polished bedrock via exposure dating is a commonly used method for studying the dynamics of glaciers and the climate changes that occurred during the) Late Pleistocene".

Lines 37-38: Change "during the Last Glacial Cycle (LGC), which spanned from approximately 120 to 11.7 ka" by "during the Late Pleistocene".

A: Change as suggested.

Line 40: Add "at regional and local scales" after "past climate variations".

A: Change as suggested.

Line 43: Change "the calculation of past environmental variability" by "to deduce past climate variability".

A: Change as suggested.

Line 46: Change "This is usually achieved through the dating of glacial landforms, such as till and polished bedrocks" by "through mapping and dating of ablation moraines".

A: This sentence has been changed as: "This is usually achieved through the geomorphological mapping and dating of till and moraines."

Lines 52-53: Chronology of the PLGM is constrained by many data from OSL datings as reported here (Lewis et al., 2009) but also by TCN datings (see Delmas et al., 2022 for review).

A: Thank you for the advice. We have added the cosmogenic dating too as: "However, during the Pyrenean Last Glacial Maximum (PLGM) period, i.e. 60-70 ka BP, corresponding to Marine Isotopic Stage (MIS) 4, few glaciers covered a significant portion of the region above 800 m a.s.l l in the Western Pyrenees, constituting a prominent element of landscape modeling in the Mediterranean area (Oliva et al., 2019), as reported by many OSL dates (Lewis et al., 2009) but also by TCN dating (see Delmas et al., 2022 for review)."

Line 53: Caution! at the time of the PLGM, only the snout of the Pyrenean glaciers reaches 800 m a.s.l. (i.e. very little surface of ice).

A: View the reply to the previous comment.

Line 55: "and throughout the LGC": Given you note above that PLGM is dated about 60-70 ka, you do not need to precise "throughout the LGC (or Late Pleistocene)

A: This part of the sentence has been removed.

Lines 55-57: "Some are better preserved as a later stage of the PLGM during MIS 4, for example in the Ara valley (Bartolomé et al., 2021) or in the Gállego valley (Lewis et al., 2009)." This sentence remains unclear, not enough accurate; reader do not understand what do you mean.

A: This sentence has been rewrite as: "For example, during the MIS 4, glacial stages are preserved in the Ara valley (Bartolomé et al., 2021) or in the Gállego valley (Lewis et al., 2009).".

Lines 62-63: "Since just few moraines were dated from this period in certain Eastern Pyrenean valleys (Noguera-Ribagorzana valley; Delmas, 2015; Pallàs et al., 2006)". Pleased, here, cite also Delmas et al. 2008, 2011, 2022 and Pallas et al., 2010, i.e. Other valley in eastern Pyrenees: Tet, Ariège, Carol, Malniu.

A: Change as suggested.

Lines 64-65: These papers do not locate the terminal position of Gallego and Cinca glaciers at the time of the global LGM, they do not indicate if they were located near or far from the PLGM.

A: This sentence has been moved, following the explanation of the early deglaciation of the Pyrenees.

Lines 81-82: "last deglaciation": I suggest you note here: "former glaciers fluctuations".

A: Change as suggested.

Line 84: 10Be dating were really sampled on boulder from moraine ridges?

A: This information has been clarified in the methodology as: "The samples included 7 granitic till boulders (PDE-1, -2, -4, -5, -6, -7, -12), 6 boulders on moraine ridges (PDE-8, -9, -10, -11, -13, -14) and one (PDE-3) from a polished quartzite bedrock".

Line 85: Change "the deglaciation of the Esera glacier" by "Late Pleistocene glacier fluctuations".

A: Change as suggested and added the Holocene as some samples date from the LIA.

Line 86: "and history of glacial advance and retreat in the study area". You do not need to repeat that given it is noted in the previous sentence.

**A: This sentence has been removed.**

Line 88: I suggest you note here "glacial stages". A glacial stage refers to the extent of a former glacier at an accurate time.

**A: Change as suggested.**

Line 125: When you sample for TCN dating, you may sample only two kinds of surfaces: boulder surface or polished surface on the bedrock. About surfaces sampled on boulders, you should precise if sampled boulders were isolated (i.e. being laid of the bedrock without any patches of till) or if they were associated to a moraine ridge. This information is very important for the interpretation of the TCN results.

A: This information has been clarified in the methodology as: "The samples included 7 granitic till boulders (PDE-1, -2, -4, -5, -6, -7, -12), 6 boulders on moraine ridges (PDE-8, -9, -10, -11, -13, -14) and one (PDE-3) from a polished quartzite bedrock".

Lines 129-130: It would be interesting and very important to locate the sampled boulders and the sampled polished surfaces on bedrock with respect to the ablation moraines ridges preserved in the catchment in order to be able to associate the TCN dates to a specific glacial stage (the same ablation moraines ridge you use to reconstruct ELA).

A: Figure 2 has been modified, differentiating the colours of the dots, depending on the type of sample recollected. As it is now, it is easier to associate the TCN dates to a specific glacial stage (also modified Table 2 to this objective).

Line 171: Change "during the LGC" by "Late Pleistocene".

**A: Change as suggested and added "Holocene".**

Lines 176-177: You should describe more accurate the geomorphological markers used to reconstruct ELA. Each ELA must correspond to a specific frontal and or lateral moraine ridge because this kind of deposit (and landform), and only this one, is able to delineate the ice margin position of a specific glacial stage. You do that accurately for LIA but not for older glacial stage. Please, do it for ALL glacial stages.

A: This information has been added as: "For each calculated ELA, the glacier surface was reconstructed, considering the frontal moraines and till previously mapped and dated with cosmogenic isotopes. To determine the lateral extension of the Ésera glacier, lateral moraines, till and in some cases very clear thresholds that avoid the expansion of very thin layer of ice previously mapped, were considered (Vidaller et al., 2024a)."

Lines 179-180: Older moraines are less preserved, but they exist. Hence, you have to tell us which moraines you used to reconstruct ELA older than LIA.

A: This issue is answered in the previous question. Fig. 2 indicates the position of the moraines older than LIA.

Lines 180-181: "the elevation of the upper limit of the frontolateral moraines from this period." This is unclear... Do you talk about "lateral moraine method" after Porter (2001)?

A: No, we talk about the Mean Equilibrium-Line Altitude method. This part has been cleared as: "Given the favorable state of preservation of the moraines from the Little Ice Age (LIA), the LIA ELAs obtained using the AABR method were compared with the LIA ELA obtained with the MELA method (Mean Equilibrium-Line Altitude; Serrano et al., 2012). This method considers that elevation of the upper limit of the frontolateral moraines coincides with the ELA of that moment, as is at this elevation where the glacier began to melt."

Line 183: You do not explain how do you constrain the ELA after orthophotos??? To constrain current ELA, you need to record data from glaciological balises on the field.

A: To clarify this a sentence has been added to the revised manuscript: "To determine the ELA in 2023 the line that separated the accumulation area (snow area) from the ablation area (ice area) was drawn manually from UAV photos, and we calculated the mean elevation of this line for the three glaciers."

Line 186-187: This suppose the older AGT (at the time of older and younger dryas) was the same than the current AGT. This is not obvious and should be discussed later in the paper.

A: It is true that the AGT could change thought time, but this is the most accurate way to determine changes in temperature along time considering the geomorphological (glacial) features. Other studies such us Serrano-González (2004) has also use this methodology. Some information has been added: "Also, the TLR present some inaccuracy, because we cannot assure that this gradient has been constant trough time as it is very dependent on atmospheric humidity. Even thought, it is the only available approximation to obtain a proxy of temperature variation between periods, as shown by Serrano-Cañadas and González-Trueba (2004) and Vidaller, (2018a)"

Serrano-Cañadas, E., & González-Trueba, J. J. (2004). El método AAR para la determinación de paleo-ELAs: Análisis metodológico y aplicación en el macizo de Valdecebollas (Cordillera Cantábrica). Cuadernos de Investigación Geográfica, 30, 7–34.

Vidaller, I. (2018). Geomorfología del macizo de Eriste: cálculo de paleoELAs y consideraciones paleoambientales. https://zaguan.unizar.es/record/77933

Lines 228-229: It would be interesting to analyse the TCN results with respect to the location within the sequence of moraines ridges (hospital moraine, plan de llanos moraine, Aiguallut moraine, etc...) because these frontal and lateral moraines ridges delineate the ice extent of the Ésera glacier for several glacial still stand stadial. TCN dates from boulders located on frontal and/or lateral moraines ridges allow these event (these glacial still stand stadial) to be dated.

A: This issue has been responded in the discussion as: "The absence of moraines in this area, with only sporadic till present, lends further credence to the hypothesis that the glacier retreated rapidly during this period, never attaining a state of equilibrium."

Lines 231-233: Hence you attribute the Llanos de Hospital stadial to the Oldest Dryas (16-15 ka).

A: Yes, we associate the Llanos de Hospital stadial to the Oldest Dryas as indicated by the TCN dating of the moraine (samples PDE-13 and 14). This sentence has been rewritten as: "The samples from the furthest downstream location (PDE-13 and -14), dated at 16.26±0.67 and

 $16.64\pm0.62$  ka and situated at Llanos del Hospital (Fig. 3), were retrieved from a moraine that was formed during the OD".

Lines 235-237: These two sentences do not allow reader to understand the chronological milestones at 13.9, 12.8 and 11 ka reported in Table 2.

A: We have rewritten the sentence as: "Eight samples situated between Pllan d'Están and Aiguallut (PDE1-7 and -12) exhibit highly similar ages, within the confines of their respective error margins (Fig. 4a). These samples, located all of them in a disperse till, represent the rapid deglaciation of the Ésera valley during the B-A period (14.6-12.9 ka). In the case of, samples PDE-10 and PDE-11, obtained from a frontal moraine, these samples were associated to an equilibrium glacial stage in the Early Holocene.".

Lines 239-240: It would be interesting to add to this sentence information about ELA (because ELA is directly controlled by climate change while glacier length is controlled by climate change AND hypsometric setting of the glacier catchment.

A: Thank you for the recommendation, but in the results section, we prefer to maintain the structure and present the results from the cosmogenic dates and then the paleoELAs results. In the introduction section some information has been added: "The ELA is highly sensitive to climatic changes that modify the extension of the accumulation and ablation zones, and also it depends on the shape and hypsometry of the glaciers (even though the evolution of the glaciers is also controlled by climate change; Quesada-Román et al., 2020; Sagredo et al., 2014; Zemp et al., 2007)). This allows the deducing past climate variability through the variations in reconstructed paleoELAs (Dahl and Nesje, 1992; Sissons and Sutherland, 1976; Sutherland, 1984)". Also, in the discussion section both results (cosmogenic dates and paleoELAs) are discussed together.

Lines 240-241: So, after your data, what is the age of Pllan d'Están moraines and Aiguallut moraines?

A: This information has been added as: "Eight samples situated between Pllan d'Están and Aiguallut (PDE1-7 and -12) exhibit highly similar ages (between 12.54±0.45 and 14.34±0.52 ka)".

Lines 243-244: After your data, what was the extent of the Ésera glacier at the time of the Younger Dryas? Given this catchment host currently the biggest Pyrenean glaciers, it would be stage if the younger dryas cooling did not trigger a glacial advance.

A: Unfortunately, we do not have this information. If there was a glacial advance in this valley during the YD, it must have been small since it did not erode the moraines and till deposited in earlier times. From Pllan d'Están proglacial sediments we infer that the Ésera glacier position was located upstream Pllan d'Están. We suspect that the glacier advance associated to the YD in this valley was a small one, probably the front would be located between the position of the B-A dates and Pllan d'Están.

Line 253: Ice margin deposits used to reconstruct ice extent at several time of the Late Pleistocene is not well explained (see comments above). Hence, we miss information to understand how you

performed to calculated ELA at 13.9; 12.8 and 11 ka. For ELA at 16 ka and at 0.4 ka, we have information to understand.

A: This information has been added in the methodology section as: "For each calculated ELA, the glacier surface was reconstructed, considering the frontal moraines and till previously mapped and dated with cosmogenic isotopes. To determine the lateral extension of the Ésera glacier for each extent lateral moraines, till and in some cases very clear thresholds that avoid the expansion of very thin layer of ice previously mapped (Vidaller et al., 2024a)."

Line 257: This approach is not explained in this paper. This said, AABR method is known to be the best one currently available to reconstruct paleoELA based on frontal and lateral moraines ridges.

A: In the methodology section there is a paragraph explaining this: "The 2023 ELA was obtained from orthophotos captured during a drone survey conducted in September 2023. To determine the ELA in 2023 the line that separated the accumulation area (snow area) from the ablation area (ice area) was draw manually and determine for the three glaciers the mean elevation of this line."

Line 260-261: "The formation of these frontolateral moraines occurred at the onset of glacial retreat, coinciding with the initial stages of the ablation zone". This is not correct and confuse.

A: This sentence has been removed and the paragraph has been modified as "Furthermore, a similar exercise can be conducted using the LIA moraines, comparing the paleoELA obtained with the MELA method and the paleoELA obtained with the AABR method which are the most well-preserved moraines in the valley. The formation of these frontolateral moraines occurred at the onset of glacial retreat, coinciding with the initial stages of the ablation zone. Consequently, their elevations should align with the paleoELA of the LIA. In this instance, when solely considering the largest glaciers (Maladeta, Aneto and Tempestades), which exhibit the most well-preserved moraines and thus provide the most robust calculations, the discrepancy between the theoretical paleoELA of the LIA and the elevation of the moraine does not exceed 50 m, thereby validating our calculations."

Lines 267-268: It would be more accurate to note "during the post-LGM deglaciation", but this imply you assume that Ésera glacier covered Pllan d'Están at the time of the Global LGM.

A: Change as suggested.

Line 271: This affirmation is not so clear when you read Vidaller et al. 2024a

A: The reference has been removed. We checked this hypothesis with the cosmogenic data.

Line 272: Chane "higher" by "lower".

A: This sentence has been changed: "The greatest temperature increase occurred during the B-A period, temperature increase 0.6°C in only 1ka (Table 2)."

Table 2: Ice extent at 16 ka is based on PDE13 and 14, i.e. Llanos de Hospital glacial still stand stage. But you do not indicate on which samples (and thus on which moraines ridges and glacial stages) you calculate ELA at 13.9, 12.8 and 11 ka. This information is essential to guarantee the reproducibility of these results.

| A. Two new rows  | have been add | ed to the table i | n order to better  | reflect that information. |
|------------------|---------------|-------------------|--------------------|---------------------------|
| A. I WO HEW TOWS | nave been add | eu to the table i | II OLGEL TO DELLEI | Terrect that information. |

| Age                    | 47 ka                            | 16 ka                  | 13.9 ka                          | 12.8 ka      | 11 ka                   | 0.4 ka                | Year 2023           |
|------------------------|----------------------------------|------------------------|----------------------------------|--------------|-------------------------|-----------------------|---------------------|
| Sample                 | Pllan d'Están
core            | PDE-13
PDE-14       | PDE-1                            | PDE-12       | PDE-10
PDE-11        | PDE-8
PDE-9        | Drone images        |
| Sample type            | Core                             | Moraine ridge boulder  | Till boulder                     | Till boulder | Moraine ridge boulder   | Moraine ridge boulder | 1                   |
| Period                 | MIS3                             | Oldest Dryas           | Onset Allerød                    | End Allerød  | Early Holocene          | Little Ice Age        | Present             |
| Phase                  | Pllan d'Están
proglacial lake | Llanos del
Hospital | Pllan d'Están
subglacial lake | Aiguallut    | Salterillo-
Barrancs | Last advance          | Very small glaciers |
| ELA (m a.s.l.)         | 2517±42                          | 2410±11                | 2519±47                          | 2645±88      | 2778±77                 | 2868±89               | 3099±140            |
| ΔT (°C)
Period-2023 | 3.1±0.2                          | 3.6±0.1                | 3.0±0.2                          | 2.4±0.5      | 1.7±0.4                 | 1.1 ±0.5              |                     |

Lines 283-284: Given the importance of data from Pllan d'Están (OSL and 14C dating from lacustrine sequence), I suggest you integrate to this paper a figure summarizing data that are used here (in this paper) to reconstruct Ésera glacier fluctuations.

A: A new figure will be included summarizing the information presented in Vidaller et al. (2024) and trying to integrate it with the new data presented in this paper. The figure will be similar to this one:

Lines 286-287: Please, do not be so allusive. Detail what you mean.

A: A reference to El Portalet lacustrine record has been added.

Lines 295-296: This is a OSL dating from fluvioglacial deposit (terrace at + 20m).

A: Thank you, we have changed it.

Lines 296: The error bar for this age is 4.5 ka and not 11 ka.

A: Change as suggested.

Line 296: This is a weiteg age from 3 OSL dating on a fluvioglacial deposit (terrace Qt7)

A: Thank you, this information has been added.

Line 297: In Pyrenees, they are other evidences of glacial advances at the beginning of the Late Pleistocene: see Pallas et al., 2010 and Delmas et al., 2011.

A: Thank you, this information has been added.

Lines 320-321: This argument is not so clearly stated in Vidallet et al., 2024b

A: As we answered in another similar question, we have deleted this reference. We checked this hypothesis with the cosmogenic data.

Line 324: You can be more affirmative given the moraine fated at 36 ka is not so far from the PLGM.

A: The word "probably" has been deleted to be more affirmative.

Line 331-332: Ok but what is your opinion about the ice extent in Ésera Valley at the time of the global LGM (did the ice covered Pllan d'Están?)

A: We have already responded to this question above, at the beginning of the letter as a "major comment". Also, it is very difficult with our data to determine the front of the Ésera glacier during the LGM but we are certain that the glacier covered Pllan d'Están as indicated the type of sediments and the fact that the Llanos del Hospital moraine is dated at 16 ka BP. The sentence has been completed as: "In the headwaters of the Ésera valley, for instance, there are no published dates that correspond to this period (23-18 ka), but surely Pllan d'Están was covered by ice, still being a subglacial lake."

Lines 334-335: You can also cite Pallas et al., 2010; Delmas et al., 2011.

A: Refences added.

Line 352: For this valley, please, cite Delmas et al., 2011 and 2012.

A: Refences added.

Line 353: For TCN dating around 16 ka in Pyrenees, you can also cite Reixach et al., 2021, Pallas et al., 2010 and Andres et al., 2018.

A: Refences added.

Lines 361-362: Hence you must assume that Pllan d'Están was covered by ice at the time of Global LGM and at the time of Oldest Dryas. In order to well understand this sentence, it is very

important to give (in this paper) a figure reporting all 14C and OSL dating (with error bar) obtained in the lacustrine sequence of Pllan d'Están.

A: A new figure (Fig 5) has been added in order to clarify this issue.

Line 402: add the reference of Reixach et al. (2021) with the reference of Jomelli et al. (2020).

A: Refence added.

Line 455: Change "deglaciation" by "ice fluctuation".

A: Change as suggested.

Andrés, N. de, Gómez-Ortiz, A., Fernández-Fernández, J.M., Tanarro, L.M., Salvador-Franch, F., Oliva, M., Palacios, D., 2018. Timing of deglaciation and rock glacier origin in the southeastern Pyrenees: a review and new data. Boreas 47, 1050–1071.

Delmas, M., Calvet, M., Gunnell, Y., Braucher, R., Bourlès, D., 2012. Les glaciations quaternaires dans les Pyrénées ariégeoises : approche historiographique, données paléogéographiques et chronologiques nouvelles. Quaternaire 23, 61–85.

Delmas, M., Gunnell, Y., Calvet, M., Reixach, T., Oliva, M., 2022. The Pyrenees: glacial landforms prior to the Last Glacial Maximum (chapter 40). In: Palacios, D., Hughes, P., García-Ruiz, J.M., Andrés, A. (Eds.), European Glacial Landscapes (volume 1): Maximum Extent of Glaciations. Elsevier, 295–307.

Delmas, M., Gunnell, Y., Calvet, M., Reixach, T., Oliva, M., 2022. The Pyrenees: glacial landforms from the Last Glacial Maximum (chapter 59). In: Palacios, D., Hughes, P., García-Ruiz, J.M., Andrés, A. (Eds.), European Glacial Landscapes (volume 1): Maximum Extent of Glaciations. Elsevier, 461–472.

Delmas, M., Gunnell, Y., Calvet, M., Reixach, T., Oliva, M., 2023. The Pyrenees: environments and landforms in the aftermath of the LGM (chapter 21). In: Palacios, D., Hughes, P., García-Ruiz, J.M., Andrés, A. (Eds.), European Glacial Landscapes (volume 2): Last Deglaciation. Elsevier, 185–200.

Delmas, M., Oliva, M., Gunnell, Y., Reixach, T., Fernandes, M., Fernández-Fernández, J.M., Calvet, M., 2023c. The Pyrenees: glacial landforms from the Younger Dryas (chapter 56). In: Palacios, D., Hughes, P., García-Ruiz, J.M., Andrés, A. (Eds.), European Glacial Landscapes (volume 2): Last Deglaciation. Elsevier 441–452.

Delmas, M., Oliva, M., Gunnell, Y., Fernández-Fernández, J.M., Reixach, T., Fernandes, M., Chapron, E., René, P., Calvet, M., 2024. The Pyrenees: glacial landforms from the Holocene (chapter 21). In: Palacios, D., Hughes, P., Jomelli, V., Tanarro, L.M., (Eds.), European Glacial Landscapes during the Holocene (volume 3). Elsevier, 419–443.

Pallàs, R., Rodés, A., Braucher, R., Bourlès, D., Delmas, M., Calvet, M., Gunnell, Y., 2010. Small, isolated glacial catchments as priority targets for cosmogenic surface exposure dating of Pleistocene climate fluctuations, southeastern Pyrenees. Geology 38, 891–894.

Reixach, T., Delmas, M., Braucher, R., Gunnell, Y., Mahé, C., Calvet, M. 2021. Climatic conditions between 19 and 12 ka in the eastern Pyrenees, and wider implications for atmospheric circulation patterns in Europe. Quaternary Science Reviews 260, 106923

I hope these comments we help you to improve the manuscript.

Magali Delmas

---

## Author Comment (AC3)

**Reviewer 2**

Review of the manuscript cp-2024-75 "Geochronological reconstruction of the glacial evolution in the Ésera valley (Central Pyrenees) during the last deglaciation" by Vidaller et al.

This manuscript deals with the reconstruction of different stages of glacier retreat and readvances (here called last deglaciation) in northern Spain. The authors use cosmo nuclide dating, as well as isotope ratios to constrain the glaciation history. The methods are robust, even though paleoELA is subject to quite some uncertainties in terms of interpretation of impacts of temperature versus precipitation changes. It would be good to defend this point in a more convincing manner — with this being said, it is a standard practice in the geomorphological community and is thus acceptable. One very interesting discovery is the subglacial paleolake that existed under the glacier during the glacier readvance prior to its final, relatively smooth retreat.

I enjoyed reading the manuscript – it is relatively well written, presents a nice story and new, previously non-existent data that are high quality and well-aligned in chronology as opposed to earlier studies in the same area characterized by a large scattering. I particularly liked the discussion in light of existing proxy data in the region and globally. Nicely done. I recommend publication after minor revisions that are listed below.

We greatly appreciate this constructive remark and concur on the significance of examining lacustrine records linked to glacier evolution, as they provide crucial evidence for understanding the timing and patterns of glacial retreat. The Pllan d'Están sequence constitutes an exemplary case study illustrating the potential of this research approach.

**General comments**

Please, edit for typos. There are quite many of them. I will pass the file with some typos marked through the editorial system.

Answer (hereinafter A): Thank you, we have reviewed all the typos.

The title with "last deglaciation" is a bit misleading because the authors also discuss the PLGM conditions at 75 ka and MIS3 evidence of glacier shrinking. I understand that it could be inferred that the last deglaciation actually started at 75 ka and continued until now but this is not commonplace to state something like this and besides the choice of a journal (CP, which is focusing on the climate reconstructions, not on paleoglaciology or glacial geomorphology as such) requires that the authors make their messages very clear and use generic language. I therefore suggest that the authors simplify their narrative for the cross-discipline communities.

A: It is true that some terms are too specific. We have tried to simplify the nomenclature using more generic language, including in the title, to make it understandable for cross-disciplinary communities.

The snow shielding effect is parameterized using quite an old methodology. I am not an expert in this subject, so I am asking whether there are any other recent publications that would introduce

a more up-to-date parameterization and how the choice of methodology impacts your conclusions. Please, elaborate and include a comparison in the appendix.

The equation from Gosse and Phillips (2001), used in this study to correct for snow cover, remains a standard approach for adjusting cosmogenic nuclide production via high-energy neutron spallation (e.g., Palacios et al., 2019; Ye et al., 2023). Recently, it has been suggested that the mass attenuation length for high-energy neutrons should be adjusted for snow, as neutron modulation by hydrogen makes difference in attenuation lengths from that in other media (Zweck et al., 2013; Delunel et al., 2014). Zweck et al. (2013) used Monte Carlo simulations to derive a lower mass attenuation length of 109 g/cm² for high-energy neutrons (100-200 MeV) in snow, which is much lower than the values for lithological media (140–170 g/cm²; Cerling and Craig, 1994).

We have included a Supplementary Table (Table S4) with age calculations for our samples, using the lower attenuation length (109 g/cm²) and comparing the results to those in the main text (Table 1). Using a lower attenuation length reduces cosmic-ray flux, leading to lower production rates and older exposure ages for a given nuclide concentration. The alternative ages calculated with the lower attenuation length are only slightly older (<1%-6%) and remain within 1-sigma errors (internal and external). The only exceptions are Samples PDE-1 and PDE-2 from valley-bottom locations, where ignoring boulder height causes an offset slightly exceeding the 1-sigma internal error but still within the 2-sigma error.

Table S4. Cosmogenic 10Be exposure ages from Ésera valley calculated using the lower attenuation length for high-energy neutrons in snow

| Field ID a | Best 10 Be age (ka) a | Int. Err | Ext. err | Comparison to the Best 10 Be age in Table 1 |
|-----------------------|---------------------------------------------|----------|----------|--------------------------------------------------------|
| PDE-1                 | 14.49                                       | 0.56     | 1.05     | 1.06                                                   |
| PDE-2                 | 13.94                                       | 0.50     | 0.99     | 1.06                                                   |
| PDE-3                 | 14.02                                       | 1.17     | 1.45     | 1.06                                                   |
| PDE-4                 | 14.34                                       | 0.52     | 1.02     | 1.00                                                   |
| PDE-5                 | 13.28                                       | 0.50     | 0.95     | 1.01                                                   |
| PDE-6                 | 12.66                                       | 0.45     | 0.90     | 1.01                                                   |
| PDE-7                 | 13.58                                       | 0.49     | 0.96     | 1.00                                                   |
| PDE-8                 | 0.433                                       | 0.037    | 0.045    | 1.01                                                   |
| PDE-9                 | 0.405                                       | 0.052    | 0.057    | 1.01                                                   |
| PDE-10                | 11.14                                       | 0.55     | 0.88     | 1.01                                                   |
| PDE-11                | 10.81                                       | 0.40     | 0.77     | 1.00                                                   |
| PDE-12                | 13.64                                       | 0.51     | 0.98     | 1.02                                                   |
| PDE-13                | 17.10                                       | 0.63     | 1.23     | 1.03                                                   |
| PDE-14                | 16.45                                       | 0.68     | 1.22     | 1.01                                                   |

Uncertainties are in one sigma: Internal (Int.) errors include only analytical errors and external (Ext.) errors include also systematic errors (such as errors associated with production rate and half-life). When compared to other geochronological data, external errors must be considered.

<sup>a Calculated assuming erosion rates at 3 mm/ka and using mass attenuation length of 109 g/cm2 (Zweck et al. 2013) for high-energy neutrons in snow for correcting for snow cover effect considering respective boulder heights, where for PDE1 and PDE2, which are located in the valley bottom, no boulder heights was considered (see text).

In conclusion, the choice of mass attenuation length for high-energy neutrons does not significantly affect our conclusions. To clarify this minor variation, we added the following sentences at the end of Section 3.1 (now in lines 186-190): "It is noted that the mass attenuation length for high-energy neutrons responsible for spallation reactions producing cosmogenic 10Be may be shorter in snow (109 g/cm²) than the conventional value assumed here (160 g/cm²), due to hydrogen moderation (Zweck et al., 2013). Calculating exposure ages with this lower value shows minimal differences for most samples (within 1-sigma errors) and only slight differences for Samples PDE-1 and PDE-2 (within 2-sigma errors; see Table S4 in the Supplementary Information)."

**References:**

- Cerling, T.E., Craig, H. (1994) Geomorphology and in-situ cosmogenic isotopes. Annual Review of Earth and Planetary Sciences, 22, 273–317.
- Delunel, R., Bourlès, D.L., van der Beek, P.A., Schlunegger, F., Leya, I., Masarik, J., Paquet, E. (2014) Snow shielding factors for cosmogenic nuclide dating inferred from long-term neutron detector monitoring. Quaternary Geochronology, 24, 16-26.
- Gosse, J., Phillips, F. (2001) Terrestrial in situ cosmogenic nuclides: theory and application. Quaternary Science Reviews, 20, 1475–1560.
- Palacios, D., Gómez-Ortiz, A., Alcalá-Reygosa, J., Andrés, N., Oliva, M., Tanarro, L.M., Salvador-Franch, F., Schimmelpfennig, I., Fernández-Fernández, J.M., Léanni, L., ASTER Team (2019) The challenging application of cosmogenic dating methods in residual glacial landforms: The case of Sierra Nevada (Spain). Geomorphology, 325, 103-118.
- Ye, S. Cuzzone, J.K., Marcott, S.A., Licciardi, J.M., Ward, D.J., Heyman, J., Quinn, D.P. (2023) A quantitative assessment of snow shielding effects on surface exposure dating from a western North American 10Be data compilation. Quaternary Geochronology, 76, 101440.
- Zweck, C., Zreda, M., Desilets, D. (2013) Snow shielding factors for cosmogenic nuclide dating inferred from Monte Carlo neutron transport simulations. Earth and Planetary Science Letters, 379, 64-71.

Again, for PaleoELA calculations – a very old publication. I understand that everyone is using ELA approach but the authors could at least add a discussion of how alternative methods could differ in terms of climate interpretation and how robust it is.

A: It is true that the references mentioned in the text are a bit old, but there are not any recent study defending the use of a new method, all the recent investigations use the same method (AABR), considering its limitations. In this sense, a new paragraph was added in order to clarify this issue (lines 54-59 in the revised manuscript): "There are several methods to determine the ELA for a glacier, but the most accurate method is the Area x Altitude Balance Ratio (AABR; Benn et al., 2005), although this method presents challenges associated with the reconstruction of past glacier extensions, particularly the identification and dating of geomorphological features that are not always well preserved to reconstruct the surface of glaciers, overthought this

approach remains a valuable method for understanding past thermal changes in mountainous environments (Pellitero et al., 2019)."

The glacier has been retreating since the LIA. Why didn't you use more robust, recent observations to validate your ELA calculations? There are plenty of remote-sensed studies, aerial surveys, and maybe even some repeat photography. Besides, there are climate reanalysis data and observations to validate your choices of parameters.

A: Since there are no remote-sense studies or aerial surveys for the LIA, only some pictures and old photos from the first alpinists are available. Even thought, there are very well conserved moraines for this cold period, so the best option is to reconstruct the glacier surface using the glacial geomorphology feature. A sentence was added (now in lines 217-219): "Unfortunately, drawings or photographs of enough quality for the LIA period to better constrain the ELA are not available, so the best option was to use the glacial geomorphology features to reconstruct the glacier surface." For the current situation, we have used more robust analyses and observations published in our previous papers that are referenced in the main text (Vidaller et al., 2021, 2023).

In the line 255 you mention equations. What equations are we talking about? There are no eq. numbers, neither do I find many equations in this work. Namely it is just one it seems. Please, clarify.

A: This sentence has been changed as (now in lines 298-299): "This has been achieved considering the ELA obtained with the AABR method for each moment and the TLR calculated with eq. 1"

Temperature lapse rate does not only depend on elevation but also on the season (can go from 4.5C/km in summer to 10C/km in winter) and even on the background climate. Please, add it in your discussion of limitations. I think such a section might need to be introduced in the appendix. Since this is part of a PhD study, such section is anyway needed in the thesis.

A: For this study mean annual temperature has been used, but as R1 and R2 the temperature lapse could vary between different periods. A new sentence has been added in the review manuscript to mark the limitations of the method (now in lines 481-486): "PaleoELAs have been demonstrated to serve as effective proxies in the determination of temperature variations during periods for which instrumental records are unavailable. However, it should be noted that the method is not without its limitations. In certain instances, such as the present case, the temperature variation values obtained through the utilisation of paleoELAs do not exactly correspond with those obtained through the application of alternative proxies. Among other limitations in the method, we are aware of the likely changes in the temperature lapse rate with elevation and with the season throughout the different considered periods."

**Minor comments:**

Line 260: Not obvious that the formation of moraines is marking the onset of glacier retreat. Aren't they accumulate during stillstands? Also, "ablation zone" seems out of context here. Further discussion/explanations are needed here.

A: This sentence has been removed.

Line 272: I don't understand these calculations. Please, explain in a better way.

A: The variation of temperature is based on the variation of the paleoELA. Considering a stable AGT throughout time, the difference of temperature is based on the difference of the elevation of the paleoELAs. This sentence has been changed as (now in lines 298-299): "This has been achieved considering the ELA obtained with the AABR method for each moment and the TLR calculated with eq. 1"

Table 2: Correct "Period - 2023". It is the other way around. Clarify if it is a summer temperature difference or mean annual?

A: This sentence has been added to the table description (lines 326-328 in the revised manuscript): "Each dated age corresponds to a specific moment during a climatic period, which does not imply that there were other situations during that period that were not recorded. The TLR was calculated considering the mean annual temperature of 2020 and 2022."

In section 5.1 you use a sediment core record dated by the OSL method. Was it not possible to cross-correlate temperature anomalies?

A: If we understand correctly, you ask to correlate temperature obtained with paleoELAs and with sediment data. In this case we did not inferred temperature data of the sediment core, only the type of ambient (e.g. a forest surrounding the lake or cold climate that avoid the development of organic matter correctly). Obtaining quantitative temperatures from the Pllan d'Están sediments would have implied a more detailed palynological study or the application of biomarkers proxies.

The statements about the early PLGM in section 5.1 are not inclusive. For example, the Barents-Kara Sea ice sheet reached its local LGM during MIS5. A lot of glaciers in Asia did too. The Patagonian ice sheet did so during MIS3, as well as glaciers in New Zealand.

A: It is right, this statement refers to a European context. This has been clarified in the reviewed text (now in lines 351-354): "Therefore, the last glacial maximum extent in the Pyrenees does not correspond in time with the global Last Glacial Maximum (LGM) considering a European context, which is estimated to have occurred between 23 and 18 ka, implying the greatest advance in European glaciers, erasing the glaciological record of previous periods (Cutler et al., 2003; Toucanne et al., 2023)."

Line 325: Misleading sentence – In many parts of the world glaciers during the Penultimate LGM were more extensive. Indicate that this is only for the last glacial period.

A: Change as suggested.

Lines 356-357: Why is that? How do you justify that your choice of parameters is more reliable? Uncertainties in delta T estimates should be explicitly discussed.

A: Because our margin of error in dating is smaller and because in almost all locations we take samples from two nearby blocks and the results are very similar. In previous studies, not only is

the margin of error in dating greater, but also, in some cases, the dates of nearby blocks are very different.

Line 403: I am not sure I understand this sentence. Rock or debris-covered glaciers or both?

A: Both, this has been clarified.

Lines 408 - 411: How do you explain such a disparity? What about the role of precipitation change?

A: As in a previous comment we have answered, the method of the proxy has limitations, this is discussed in lines 353-357 of the revised manuscript.

Line 420: The formulation is weird.

A: This sentence has been rewritten as (now in lines 493-495): "Thus, some polished bedrocks at an elevation of 2549-2719 m a.s.l in the Gállego valley (Central Pyrenees), were dated with 10Be and 36Cl exposure ages, resulting in a date of 10.6±1.3 ka (Palacios et al., 2017a)."

Line 431: I agree regarding biological indicators but isn't it the same for ELA to some extent?

A: I agree with you, the ELA determines the elevation of the iso 0°C during the ablation period, so during summer. In this sense this discussion has been deleted.

Lines 439-440: Why so?

A: These values were determined in other studies.

Line 462: What mechanism can explain such an early local LGM?

A: In this sense, in the rest of Europe, there was also a glaciation around 70-60 ka, but there during the LGM (23-18 ka) glacier advanced more than during the 70-60 ka expansion. In the Pyrenees, during the LGM the weather was cold but arid, avoiding large accumulations of snow, and consequently limiting the expansion of glaciers. This is up to now the more accepted hypothesis but the lack of LGM advances in Central and Western Pyrenees is still a matter of debate in our community.

How reliable are reconstructions of glacier retreat followed by a readvance? What makes your methodology trustworthy? Explain for paleoclimatologists.

The main strength of this study is the combination of a continuous paleoclimate record, the lacustrine sequence in Pllan d'Estàn, and the discontinuous information obtained throughout the cosmo dating. Such an approach allows us to infer different episodes of glacier advances and retreats based on the combination of different type of information. For example, for the LGM we can infer that the glacier covered Pllan d'Estàn using the type of sediments accumulated during that time period in spite we don't have any moraine associated to that age. On the other hand, we can infer a glacier readvance at 16 ka BP using the Llanos del Hospital moraine dated at the Oldest

Dryas period. Ideally, the complementary study of other continuous sedimentary sequences along the Ésera valley would provide additional information to support our interpretations.